# The cell proliferation antigen Ki-67 organises heterochromatin

Michal Sobecki[1,2†], Karim Mrouj[1,2], Alain Camasses[1,2], Nikolaos Parisis[1,2], Emilien Nicolas[3], David Llères[1,2], François Gerbe[2,4,5], Susana Prieto[1,2], Liliana Krasinska[1,2], Alexandre David[2,4,5], Manuel Eguren[6], Marie-Christine Birling[7], Serge Urbach[2,4,5,8], Sonia Hem[9], Jérôme Déjardin[2,10], Marcos Malumbres[6], Philippe Jay[2,4,5], Vjekoslav Dulic[1,2], Denis LJ Lafontaine[3], Robert Feil[1,2], Daniel Fisher[1,2]*

[1]Montpellier Institute of Molecular Genetics (IGMM) CNRS UMR 5535, Centre National de la Recherche Scientifique (CNRS), Montpellier, France; [2]Faculty of Sciences, University of Montpellier, Montpellier, France; [3]RNA Molecular Biology, Center for Microscopy and Molecular Imaging, Fonds de la Recherche Nationale, Université Libre de Bruxelles, Charleroi-Gosselies, Belgium; [4]Institute of Functional Genomics (IGF), CNRS UMR 5203, Centre National de la Recherche Scientifique (CNRS), Montpellier, France; [5]U1191, Inserm, Montpellier, France; [6]Spanish National Cancer Research Centre, Madrid, Spain; [7]ICS, Mouse Clinical Institute, Illkirch-Graffenstaden, France; [8]Functional Proteomics Platform, Institute of Functional Genomics, Montpellier, France; [9]Mass Spectrometry Platform MSPP, SupAgro, Montpellier, France; [10]Institute of Human Genetics (IGH) CNRS UPR 1142, Centre National de la Recherche Scientifique, Montpellier, France

*For correspondence: daniel.fisher@igmm.cnrs.fr

Present address: [†]Department of Genome Biology, Institute for Integrative Biology of the Cell, Université Paris Sud, Gif-sur-Yvette, France

Competing interests: The authors declare that no competing interests exist.

**Abstract** Antigen Ki-67 is a nuclear protein expressed in proliferating mammalian cells. It is widely used in cancer histopathology but its functions remain unclear. Here, we show that Ki-67 controls heterochromatin organisation. Altering Ki-67 expression levels did not significantly affect cell proliferation in vivo. Ki-67 mutant mice developed normally and cells lacking Ki-67 proliferated efficiently. Conversely, upregulation of Ki-67 expression in differentiated tissues did not prevent cell cycle arrest. Ki-67 interactors included proteins involved in nucleolar processes and chromatin regulators. Ki-67 depletion disrupted nucleologenesis but did not inhibit pre-rRNA processing. In contrast, it altered gene expression. Ki-67 silencing also had wide-ranging effects on chromatin organisation, disrupting heterochromatin compaction and long-range genomic interactions. Trimethylation of histone H3K9 and H4K20 was relocalised within the nucleus. Finally, overexpression of human or *Xenopus* Ki-67 induced ectopic heterochromatin formation. Altogether, our results suggest that Ki-67 expression in proliferating cells spatially organises heterochromatin, thereby controlling gene expression.

## Introduction

The cell proliferation antigen Ki-67 (Ki-67 or Ki67) is constitutively expressed in cycling mammalian cells (*Gerdes et al., 1983*). It is therefore widely used as a cell proliferation marker to grade tumours. Ki-67 is a nuclear DNA-binding protein (*MacCallum and Hall, 2000*) with two human isoforms that have predicted molecular weights of 320kDa and 359kDa (*Gerdes et al., 1991*). The domain structure of Ki-67 is represented in *Figure 1*. All homologues contain an N-terminal Fork-head-associated (FHA) domain, which can bind both to DNA and to phosphorylated epitopes. The

**eLife digest** Living cells divide in two to produce new cells. In mammals, cell division is strictly controlled so that only certain groups of cells in the body are actively dividing at any time. However, some cells may escape these controls so that they divide rapidly and form tumors.

A protein called Ki-67 is only produced in actively dividing cells, where it is located in the nucleus – the structure that contains most of the cell's DNA. Researchers often use Ki-67 as a marker to identify which cells are actively dividing in tissue samples from cancer patients, and previous studies indicated that Ki-67 is needed for cells to divide. However, the exact role of this protein was not clear. Before cells can divide they need to make large amounts of new proteins using molecular machines called ribosomes and it has been suggested that Ki-67 helps to produce ribosomes.

Now, Sobecki et al. used genetic techniques to study the role of Ki-67 in mice. The experiments show that Ki-67 is not required for cells to divide in the laboratory or to make ribosomes. Instead, Ki-67 alters the way that DNA is packaged in the nucleus. Loss of Ki-67 from mice cells resulted in DNA becoming less compact, which in turn altered the activity of genes in those cells.

Sobecki et al. also identified many other proteins that interact with Ki-67, so the next step following on from this research is to understand how Ki-67 alters DNA packaging at the molecular level. Another future challenge will be to find out if inhibiting the activity of Ki-67 can hinder the growth of cancer cells.

most characteristic feature of Ki-67 is the presence of multiple tandem repeats (14 in mice, 16 in human) containing a conserved motif of unknown function, the 'Ki-67 domain'. Two other conserved motifs include a Protein Phosphatase 1 (PP1)-binding motif (*Booth et al., 2014*) and a 31 amino acid conserved domain (CD) of unknown function, 100% identical between human and mouse, that includes a 22 amino acid motif conserved in all homologues. Ki-67 homologues also have a weakly conserved leucine/arginine rich C-terminus which can bind to DNA and, when overexpressed, promotes chromatin compaction (*Scholzen et al., 2002*; *Takagi et al., 1999*).

Ki-67 protein levels and localisation vary through the cell cycle. Its maximum expression is found in G2 phase or during mitosis (*Endl and Gerdes, 2000b*). In interphase, Ki-67 forms fibre-like structures in fibrillarin-deficient regions surrounding nucleoli (*Verheijen et al., 1989b*; *Kill, 1996*; *Cheutin et al., 2003*). Ki-67 also colocalises with satellite DNA (*Bridger et al., 1998*) and is found in protein complexes that bind to satellite DNA (*Saksouk et al., 2014*). It remains associated with nucleolar organiser regions of acrocentric chromosomes throughout interphase (*Bridger et al., 1998*). Ki-67 is a direct substrate of the cyclin-dependent kinase CDK1 (*Blethrow et al., 2008*) and is hyperphosphorylated in mitosis. This may regulate its expression and / or localisation (*Endl and Gerdes, 2000a*). In HeLa cells, Ki-67 binds tightly to chromatin in interphase, whereas this binding is weakened in mitosis (*Saiwaki et al., 2005*) when it associates with condensed chromosomes before relocating to the chromosome periphery (*Verheijen et al., 1989a*).

Ki-67 is generally assumed to be required for cell proliferation. An early study found that injection of antisense oligonucleotides that block Ki-67 expression inhibited cell proliferation (*Schluter et al., 1993*). Subsequent studies have shown that unperturbed Ki-67 expression levels are required for normal proliferation rates in various cell lines (*Kausch et al., 2003*; *Rahmanzadeh et al., 2007*; *Zheng et al., 2006*; *2009*). However, no complete Ki-67 loss of function studies have been reported. Therefore, it remains unclear what its functions are and whether it is essential for cell proliferation.

The dynamic localisation of Ki-67 has led to suggestions that it could coordinate nucleolar disassembly and reassembly at either side of mitosis (*Schmidt et al., 2003*). Indeed, Ki-67 is required to localise nucleolar granular components to mitotic chromosomes, thereby potentially playing a role in nucleolar segregation between daughter cells (*Booth et al., 2014*). It has also been reported that Ki-67 has roles in ribosome biogenesis (*Rahmanzadeh et al., 2007*), consistent with its nucleolar localisation and apparent role in cell proliferation.

In this work, we characterise the cellular roles of Ki-67 using knockdown and genetic approaches. We find that mutant mice with disrupted Ki-67 expression are viable and fertile. Preventing Ki-67 downregulation upon cell cycle exit in vivo does not impede differentiation. Thus, Ki-67 expression

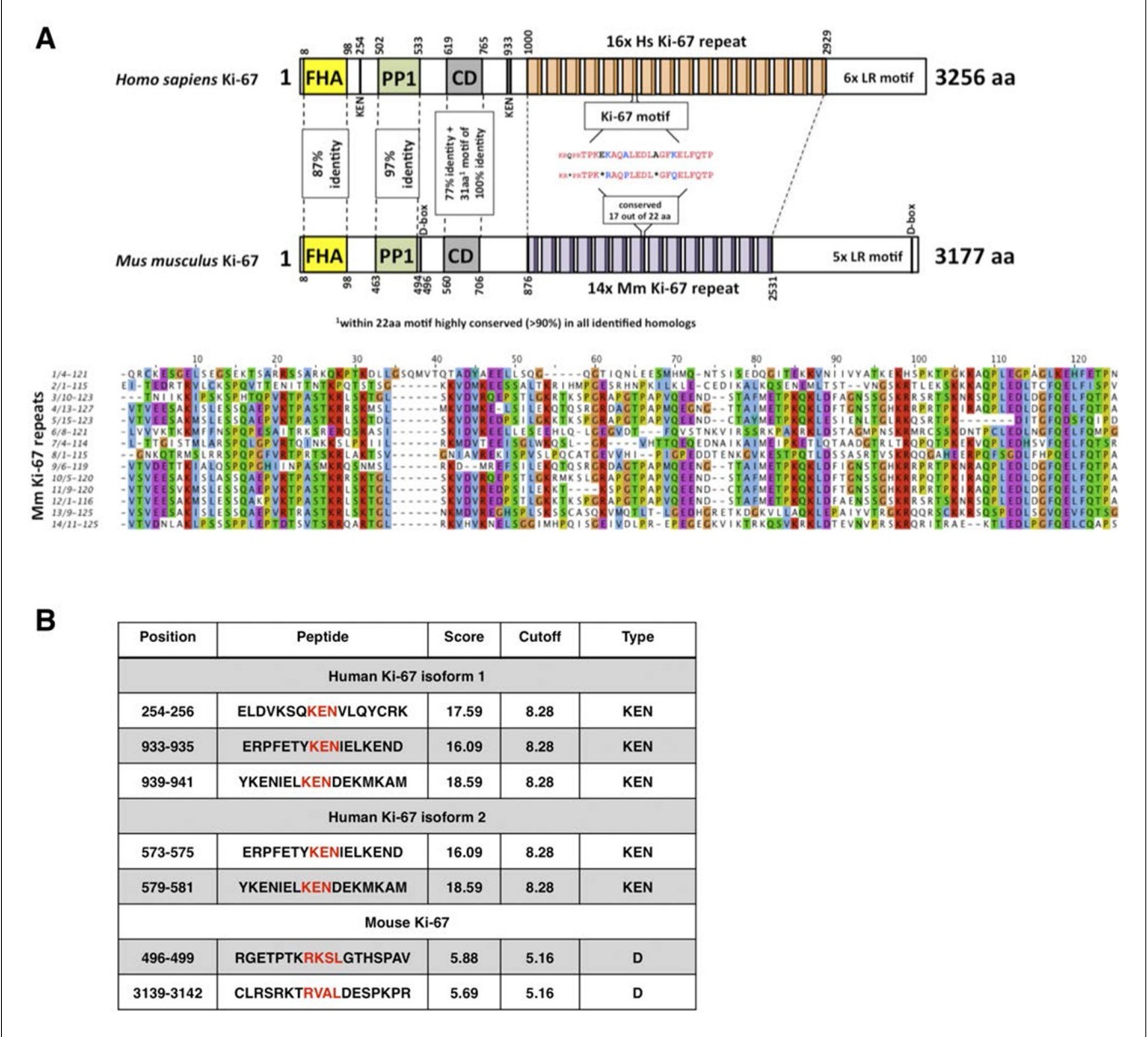

**Figure 1.** Comparison of human and mouse Ki-67 structural elements. (A) Top: cartoon of human (long form) and mouse Ki-67 protein highlighting conserved elements and functional motifs. Domains are indicated by boxes (FHA, forkhead-associated domain; PP1, PP1-binding domain; CD, conserved domain; D-box: APC/C targeting destruction box motifs; KEN: APC/C-Cdh1 targeting KEN box motifs). Highly conserved regions are indicated by dotted line with percent of identical amino acids. Bottom: alignment of mouse Ki-67 repeats. (B) APC/C targeting motifs identified in human (both isoforms) and mouse Ki-67.

can be uncoupled from cell proliferation. Instead, we show that Ki-67 is an essential mediator of heterochromatin organisation and long-range chromatin interactions, controlling gene expression. As it is expressed at high levels only in proliferating cells, our results suggest that Ki-67 links heterochromatin organisation to cell proliferation.

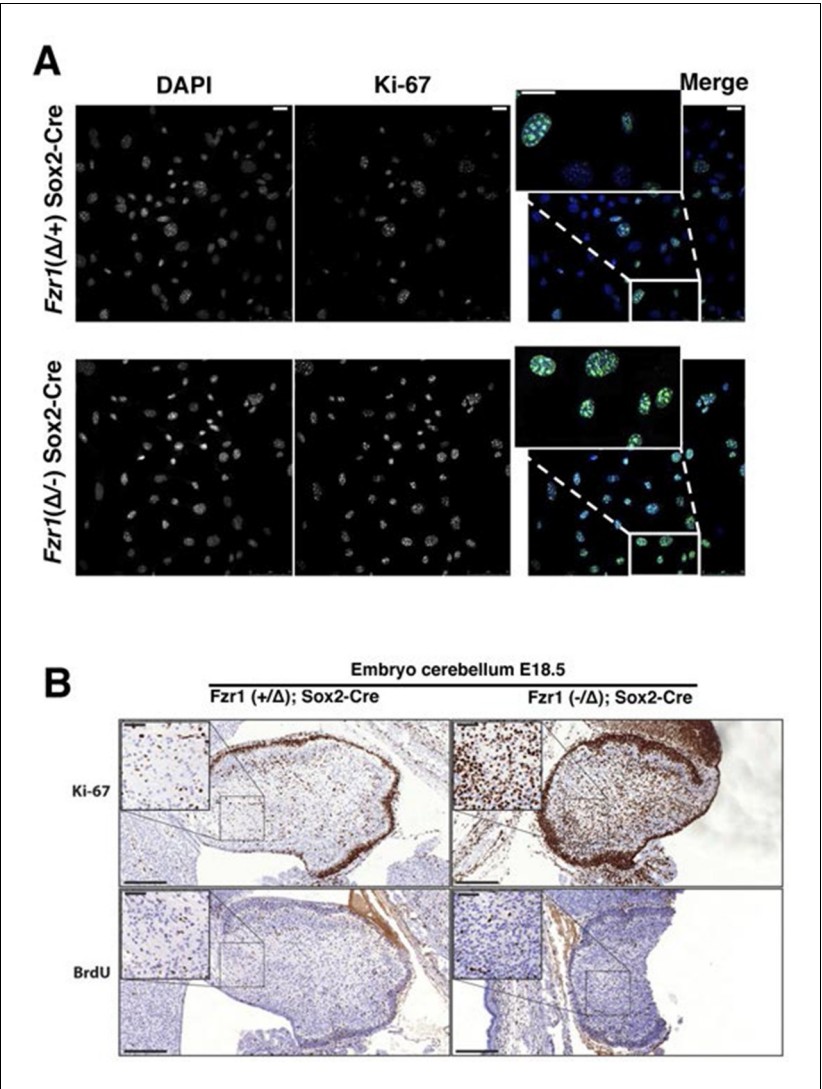

**Figure 2.** Maintenance of Ki-67 expression in quiescent cells in vivo by Cdh1 mutation. (**A**) Immunofluorescence analysis of Ki-67 in MEF cells isolated from embryo (E13.5) of *Fzr1*(+/Δ);*Sox2*-Cre and Fzr1(-/Δ);*Sox2*-Cre mice. Scale bar, 25 µm. (**B**) IHC staining of Ki-67 and BrdU in sagittal sections of embryo cerebellum (E18.5) of *Fzr1* (+/Δ); *Sox2*-Cre and *Fzr1* (-/Δ);*Sox2*-Cre mice. Bars, 200 µm. Bars in zoom, 50 µm.

The following figure supplement is available for figure 2:

**Figure supplement 1.** Ki-67 expression is restricted to proliferating cells by APC/C-Cdh1.

## Results

### Mouse development is not affected by genetic up- and downregulation of Ki-67 expression

Given the tight correlation between Ki-67 expression and cell proliferation, it is often assumed that Ki-67 is required for cell proliferation and that its downregulation might promote cell cycle exit. We tested these hypotheses genetically. Ki-67 protein expression is regulated during the cell cycle, and we speculated that it might be a target for the APC/C-Cdh1 ubiquitin ligase complex. This complex is active in late mitosis and G1, and triggers degradation of substrates containing D-boxes and KEN boxes. Human Ki-67 isoforms contain two or three KEN boxes, whereas mouse Ki-67 contains two D-boxes (*Figure 1A,B*). Mouse Ki-67 contains an additional sequence, AQRKQPSR at 2680–2687,

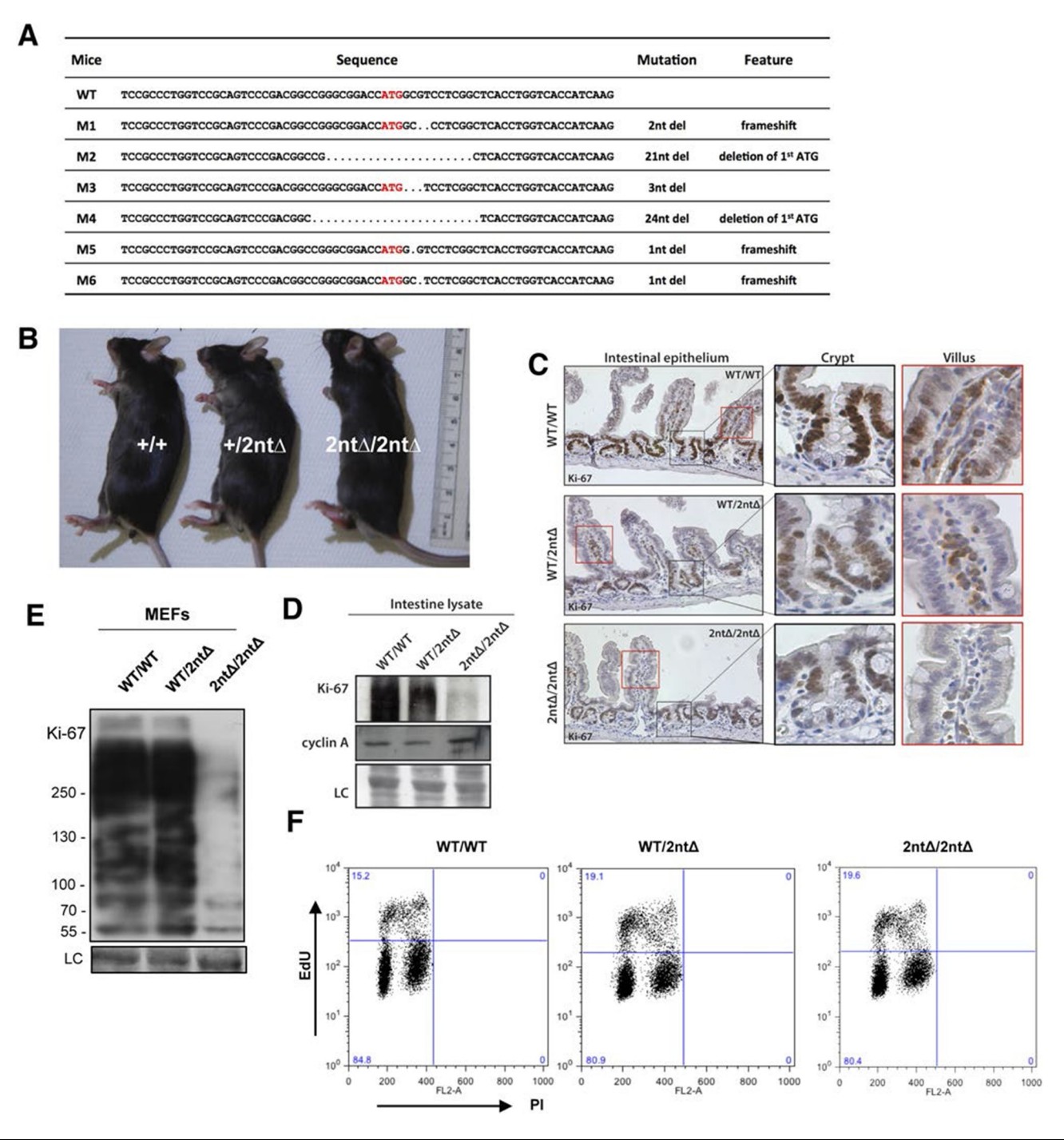

**Figure 3.** Mouse development with a mutated Ki-67 gene. (A) Table describing Ki-67 mutant mouse lines resulting from germline transmission of mutations generated by cytoplasmic injection of TALEN-encoding mRNA into zygotes. (B) Macroscopic appearance of littermate female mice at 10 weeks of age. Genotypes are specified. (C) IHC staining of Ki-67 in sagittal section of intestine from $Mki67^{WT/WT}$, $Mki67^{WT/2nt\Delta}$ and Mki67$^{2nt\Delta/2nt\Delta}$ mice. (D) Western blots of Ki-67 and cyclin A expression from intestine isolated from $Mki67^{WT/WT}$, $Mki67^{WT/2nt\Delta}$ and $Mki67^{2nt\Delta/2nt\Delta}$ mice. LC, loading control. (E) Western blot of Ki-67 in MEFs from WT, $Mki67^{WT/2nt\Delta}$ and $Mki67^{2nt\Delta/2nt\Delta}$ mice. LC, loading control. (F) Flow cytometry profiles in WT, $Mki67^{WT/2nt\Delta}$ and $Mki67^{2nt\Delta/2nt\Delta}$ MEFs showing EdU incorporation upon a 1 hr pulse and DNA content.

The following source data and figure supplements are available for figure 3:

**Figure supplement 1.** Ki-67 mutant mice develop normally.

*Figure 3 continued on next page*

*Figure 3 continued*

which is highly similar to the A-box, a third APC/C-Cdh1 recognition motif (*Littlepage and Ruderman, 2002*). To see whether Cdh1 regulates Ki-67, we analysed mouse embryo fibroblasts (MEFs) lacking the *Fzr1* gene that encodes Cdh1 (*Garcia-Higuera et al., 2008*). Asynchronous *Fzr1* heterozygous MEFs, that are at different stages of the cell cycle, had variable Ki-67 levels, whereas in the *Fzr1* knockout MEFs Ki-67 was upregulated and more homogeneously expressed (*Figure 2A*). To see whether sustained Ki-67 expression in quiescent cells would have a negative impact on cell cycle arrest in vivo, we analysed *Fzr1*-knockout mice. Here, Ki-67 was overexpressed and uncoupled from cells that incorporate BrdU in all tissues examined (*Figure 2B*, *Figure 2—figure supplement 1*). Thus, Ki-67 expression is regulated by APC/C-Cdh1 in mice and its downregulation is not a prerequisite for cell cycle exit.

We next investigated the functional consequences of Ki-67 downregulation for normal tissue development and homeostasis. To disrupt the gene encoding Ki-67, *Mki67*, in the mouse germline, we used a TALEN pair targeting the unique ATG start codon. This is predicted to generate null alleles (*Figure 3—figure supplement 1A*). After cytoplasmic injection of these TALEN-encoding mRNAs into zygotes, 10 out of 54 mice had mutations disrupting the coding sequence. We crossed founder mutant mice, and four gave germline transmission of the mutation. Due to mosaicism, this resulted in six lines: two lines with mutations eliminating the initiation codon and four lines with deletions which cause frameshifts immediately downstream of the ATG (*Figure 3A*). From these, we selected a 2-nucleotide deletion (2ntΔ) mutant that retains the ATG initiation codon but has a frameshift in the next codon (*Figure 3—figure supplement 1B*), and a 21-nucleotide deletion (21ntΔ) that eliminates the ATG (*Figure 3—figure supplement 1C*). We crossed these mice and, unexpectedly, obtained homozygous mutants at the expected Mendelian frequency that were indistinguishable from heterozygous or wild-type (WT) littermates (*Figure 3B*, *Figure 3—figure supplement 1D*). Both deletion mutant lines showed normal growth and were fertile. Sagittal sections from *Mki67*$^{2ntΔ/2ntΔ}$ mice did not reveal any obvious defects in tissue morphology (*Figure 3—figure supplement 2*). Since the intestinal epithelium is the most highly proliferative adult mouse tissue, we compared its morphology between WT and mutant mice. In WT animals, the proliferative crypt compartment was strongly stained for Ki-67 by immunohistochemistry (IHC), while only minimal levels of Ki-67 were observed in the differentiated cells on the villus (*Figure 3C*, top), as expected. In contrast, in the mutants, proliferating crypt cells showed only residual levels of Ki-67 staining by IHC (*Figure 3C*, bottom) or immunofluorescence (*Figure 3—figure supplement 3*). Immunoblotting of intestinal

epithelium preparations could detect a weak band of similar size to WT Ki-67 (*Figure 3D*; *Figure 3—figure supplement 4*). The signal was, however, reduced by at least 90% in both mutants compared to WT tissue. Three different Ki-67 antibodies gave similar results. These are all extremely sensitive as they recognise the highly repeated Ki-67 domain. They should also detect N-terminally truncated Ki-67 that would result from translation from the ATG at position 433. qRT-PCR analysis showed that Ki-67 mRNA level was, unexpectedly, increased rather than reduced in the intestinal tissue (*Figure 3—figure supplement 5*). In the intestinal epithelium, analysis of Wnt signalling and differentiation of goblet and tuft cells showed no differences between WT and *Mki67*$^{21nt\Delta/21nt\Delta}$ mice (*Figure 3—figure supplement 6*). These results show that high Ki-67 levels and an intact Ki-67 gene are not required for development or differentiation in vivo.

To see if cells from Ki-67 mutant mice had normal proliferation capacity we isolated embryonic fibroblasts (MEFs) from day-13 embryos. Homozygous *Mki67*$^{2nt\Delta/2nt\Delta}$ MEFs had at least 90% lower Ki-67 levels (*Figure 3E*). We could not confirm by immunoblotting whether or not the protein was full-length or truncated since SDS-PAGE cannot resolve 15kDa differences between proteins of nearly 400 kDa, and no antibodies are available against the N-terminus of Ki-67. As with intestinal tissue, the loss of Ki-67 expression was not due to mRNA degradation, as shown by qRT-PCR (*Figure 3—figure supplement 7*). Indeed, in the homozygous mutant, the Ki-67 mRNA level was increased to a level comparable with that of proliferating NIH-3T3 cells. Mutant MEF proliferated comparably to controls, and flow cytometric assessment of EdU incorporation after a 1 hr EdU pulse showed similar numbers of replicating cells in Ki-67 WT and mutant cells (*Figure 3F*).

The low level residual Ki-67 expression in homozygous Ki-67 mutants suggests that TALEN or the conceptually-related CRISPR approaches may not lead to complete loss of expression, even when the translation initiation codon has been mutated. To further investigate whether Ki-67 translation can occur with a mutated initiation codon, we used the same TALEN pair to generate monoclonal Ki-67 mutant mouse NIH-3T3 cell lines, allowing analyses of translation that are technically impossible using animal tissues. We also performed the same procedure in the absence of TALENs to isolate wild-type clones. We obtained nine mutants with very low Ki-67 expression. Cloning and sequencing showed that five had biallelic mutations around the ATG codon (*Figure 3—figure supplement 8*). As in mice, even though Ki-67 was visible by immunofluorescence, Ki-67 was barely detectable by Western blot in all clones analysed (*Figure 3—figure supplement 9A*). All clones proliferated efficiently (*Figure 3—figure supplement 9B*). qRT-PCR showed that mutants did not have decreased Ki-67 mRNA levels compared to WT NIH-3T3 cells; indeed, like mutant MEFs, clone 14 had a higher level (*Figure 3—figure supplement 9C*). We selected two clones for further analysis of Ki-67 translation. Clone 14 had lost the ATG codon in one allele, but had acquired an insertion of 4 nucleotides after the ATG in the second allele, generating the same shifted reading frame as the *Mki67*$^{2nt\Delta/2nt\Delta}$ mice. Clone 21 had lost the ATG codon in both alleles, thus mimicking the situation with *Mki67*$^{21nt\Delta/21nt\Delta}$ mice. qRT-PCR quantification of Ki-67 mRNA from ribosome purifications shows that in clones 14 and 21 there was no decrease in ribosome association with Ki-67 mRNA; indeed, in clone 14 it increased, probably due to the higher Ki-67 mRNA level (*Figure 3—figure supplement 10*). To definitively determine levels of Ki-67 translation in the mutants, we performed SILAC quantitative mass spectrometry from exponentially growing WT or mutant NIH-3T3 cells cultured in light (L) (WT) or heavy-labelled (H) medium (clones 14 and 21). Chromatin was purified and run on SDS-PAGE. Peptides were purified from two gel slices, one around the predicted size of full length Ki-67 (>250 kDa; band 1) and one at a smaller size (130k Da-250 kDa, band 2). Ki-67 could not be positively identified in clones 14 and 21 (*Figure 3—figure supplement 11*). In peptides from WT cells (L), 44 peptides derived from Ki-67 were identified by MS/MS. In contrast, in peptides from mutant cell lines (H), no MS/MS spectra for Ki-67 could be identified in either band. Selecting the 're-quantify' option in MaxQuant (that forces quantitation of identified light peaks against any peaks that have the expected difference in m/z ratio), the ratios H/L observed for putative Ki-67 peaks were in the range of most typical contaminants that are only found unlabelled in a SILAC experiment. In band 1, the median normalised H/L intensity ratio of the 5 corresponding m/z peaks in clone 14 was 0.095 (mean, 0.100, SD, 0.02), and in clone 21 (7 peptides) was 0.191 (mean 0.203, SD 0.05). In band 2, which would result from truncated or degraded Ki-67, there were only 3 corresponding peaks, with median H/L ratios for clones 14 and 21 of 0.358 (mean, 0.402, SD, 0.30) and 0.500 (mean, 0.454, SD 0.33) respectively. Taken together, these results show that if Ki67 is translated in the mutant cell lines, it is at trace levels that are not positively identifiable by state-of-the-art mass

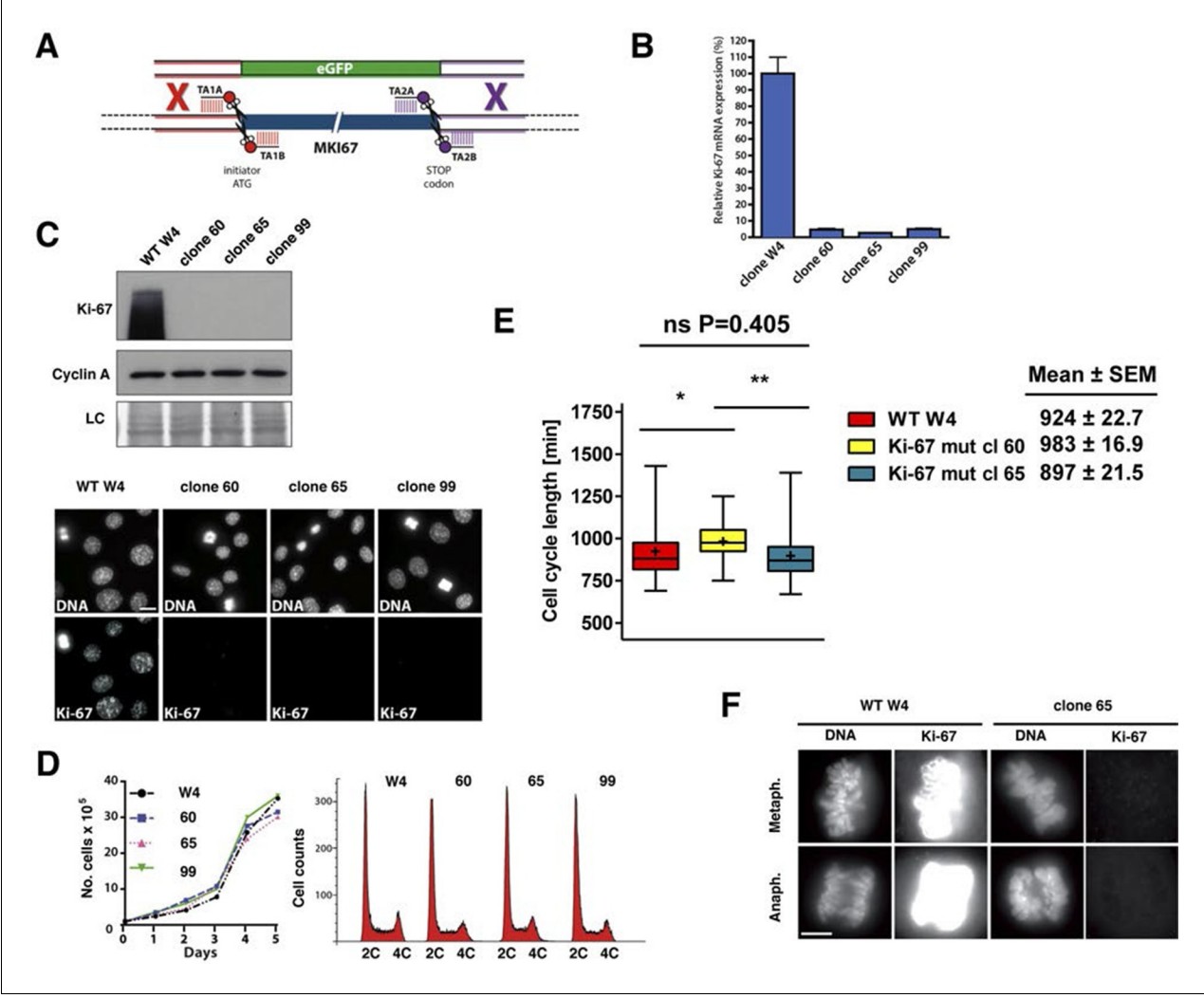

**Figure 4.** Cell proliferation without Ki-67. (**A**) Schematic representation of strategy for TALEN-mediated generation of *Mki67* null allele. (**B**) qRT-PCR analysis of Ki-67 mRNA levels in NIH-3T3 WT clone W4 and Ki-67-negative 60, 65, 99 clones. (**C**) Top: Western blot of Ki-67 and Cyclin A in NIH-3T3 WT clone W4 and Ki-67-negative mutant clones 60, 65, 99; LC, loading control; below, Ki-67 immunofluorescence; bar, 10 μm. (**D**) Left, growth curves of WT and Ki-67 null cell lines 60, 65 and 99; right, cell cycle distribution analysed by flow cytometry. (**E**) Cell cycle length of WT clone W4 and Ki-67 null clones 60 and 65 as determined by time-lapse videomicroscopy. (**F**) Cells of clone 65 show altered chromosomal periphery in mitosis. The Ki-67 staining is deliberately overexposed to demonstrate absence of detectable Ki-67 in clone 65, even in metaphase. Bar, 5 μm.

The following figure supplement is available for figure 4:

**Figure supplement 1.** Generation of NIH-3T3 cells lacking Ki-67.

spectrometry. At best, translation can occur from the mutated Ki-67 gene with an estimated 10% efficiency. The product is most likely an N-terminally truncated protein lacking the conserved FHA-domain, arising from a downstream in-frame ATG.

## Cells lacking Ki-67 proliferate efficiently

Given the above results, it remained possible that very low levels of Ki-67 remain after *Mki67* gene mutation and that they might suffice to sustain cell proliferation. To rule out this possibility we devised a 'double TALEN' strategy to completely eliminate Ki-67 expression. We designed and synthesised an additional TALEN pair targeting a sequence downstream of the translation stop codon. We co-transfected the ATG TALEN pair and the Stop TALEN pair with a GFP knock-in construct containing homology arms (*Figure 4A*). We thus isolated several monoclonal cell lines in which Ki-67

mRNA was essentially eliminated (*Figure 4B*), indicating efficient nonsense-mediated decay (NMD). These cell lines had no residual Ki-67 protein expression (*Figure 4C*), confirming that the basal immunostaining seen in clones 14 and 21 indeed reflected trace level Ki-67 expression. We used Southern blotting of genomic DNA to characterise these alleles. The 3' end of the *Mki67* gene remained intact in all clones, while the 5' end of mutants showed a rearrangement consistent with a tandem insertion of multiple copies of the knockin construct upstream of the *Mki67* ORF (*Figure 4—figure supplement 1*). Thus, the *Mki67* gene was severely disrupted but not deleted. These Ki-67-negative cells proliferated normally. Growth curves and DNA content profiles were indistinguishable from controls (*Figure 4D*). Time-lapse videomicroscopy showed that although individual clones had slightly different cell cycle lengths, cell division time was not significantly different between WT and mutant clones (*Figure 4E*). We noticed that mitotic cells of one of the clones lacking Ki-67 had an altered chromosomal periphery (*Figure 4F*). Such a phenotype has previously been reported in Ki-67 knockdown HeLa cells (*Booth et al., 2014*). Nevertheless, these cells could divide efficiently.

Next, we tested whether *Mki67* mutant clones had altered kinetics of cell cycle exit or entry. To do this, we quantified cells that could replicate by measuring 5-ethynyl-2-deoxyuridine (EdU) incorporation into DNA. We found that 42% of WT *Mki67* cells could still incorporate some level of EdU even after 72 hr with 0.1% serum, but only 13% or 28% of mutant clones 60 or 65, respectively, could do so (*Figure 5A*). This suggested that *Mki67* disruption rendered cells slightly more sensitive to serum starvation. Upon addition of serum to quiescent cells, WT and *Mki67* mutants entered the cell cycle with similar kinetics (*Figure 5A*). Ki-67 remained completely undetectable in mutants (*Figure 5B*).

As Ki-67 is frequently used to assess proliferation in human cancer cells, we tested whether human cells lacking Ki-67 can proliferate. We generated stable human cell lines with inducible or constitutive expression of shRNA that silenced Ki-67 or a non-silencing control (*Figure 5—figure supplement 1A*, *2A*). We used the non-transformed human fibroblast cell line BJ-hTERT, and two commonly used cancer cell lines, HeLa and U2OS, which are of epithelial and mesenchymal origin, respectively. In BJ-hTERT, inducing shRNA expression largely prevented Ki-67 expression but had no detectable effect on the kinetics of entry into the cell cycle, as judged by expression of cell cycle regulators and EdU incorporation (*Figure 5—figure supplement 1B*). Further reducing residual Ki-67 levels with siRNA also did not affect cell proliferation (*Figure 5—figure supplement 1C*). Similarly, constitutive knockdown of Ki-67 in stable shRNA-expressing HeLa or U2OS cells had no effect on cell cycle distribution nor on the expression of cell cycle regulators (*Figure 5—figure supplement 2B*). Analysing single cells by immunofluorescence showed that knockdown U2OS cells with undetectable Ki-67 expression incorporated EdU in a similar manner to control cells, demonstrating efficient DNA synthesis (*Figure 5—figure supplement 2C*).

Taken together, these results show that although Ki-67 elimination might have minor effects on cell cycle exit and mitosis, mammalian cells can nevertheless proliferate efficiently in the absence of detectable Ki-67.

## Ki-67 interacts with proteins involved in nucleolar processes and chromatin regulation

To investigate possible molecular functions of Ki-67, we identified interacting proteins. To do this, we expressed FLAG-tagged versions of full-length human Ki-67 or an unrelated protein (TRIM39) in U2OS cells, and pulled down proteins from nuclear extracts with anti-FLAG antibody (*Figure 6—figure supplement 1*). These were analysed by label-free mass spectrometry. This approach identified 406 proteins specific to the Ki-67 pulldown (*Figure 6—figure supplement 2*). These included known Ki-67 partners: CDK1, an established Ki-67 kinase (*Blethrow et al., 2008*), nucleolar protein NIFK (*Takagi et al., 2001*), protein phosphatase 1 (*Booth et al., 2014*), and five subunits (HCFC1, HSPA8, MATRIN3, RBBP5 and WDR5) of a histone methylase complex that interacts with the nuclear receptor coregulator NRC (also known as NCOA6; *Garapaty et al., 2009*). Gene ontology and STRING analysis classified the Ki-67 interactors as being enriched in two general processes: chromatin regulation and ribosomal biogenesis (*Figure 6*, *Figure 6—figure supplement 2*). Specifically, interactors could be subdivided into groups involved in chromatin modification and transcription, ribosomal subunit biogenesis, pre-rRNA processing, protein translation, and splicing. These interactions suggested that Ki-67 might be involved not only in ribosomal biogenesis, as previously suggested (*Rahmanzadeh et al., 2007*), but also in regulating chromatin. Among chromatin regulators

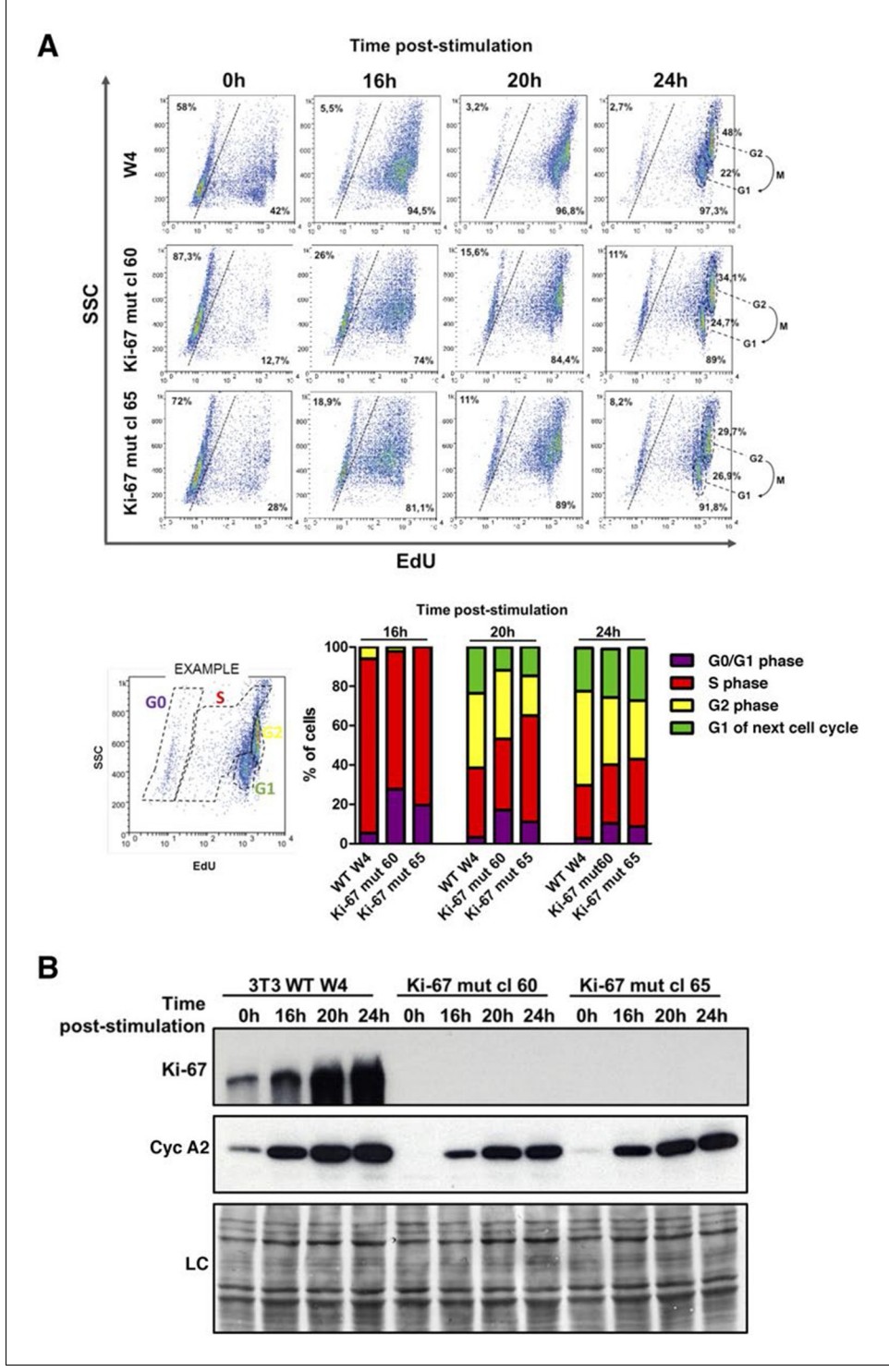

**Figure 5.** Cells lacking Ki-67 enter the cell cycle efficiently. (**A**) Top, re-entry of cell cycle in NIH-3T3 WT clone W4 and Ki-67-negative mutant clones 60 and 65 after serum starvation-induced cell cycle arrest. Progression of cell cycle entry analysed by FACS using EdU staining. Bottom, quantification of cell cycle phases in this experiment. (**B**) Western blot analysis of Ki-67 (upper panel) and cyclin A2 (lower panel) upon cell cycle entry. LC, loading control.

The following figure supplements are available for figure 5:

**Figure supplement 1.** Cells lacking Ki-67 proliferate efficiently.

*Figure 5 continued*

**Figure supplement 2.** Cells lacking Ki-67 proliferate efficiently.

interacting with Ki-67, we found the NRC-interacting methylase complex; KMT2D, ASH2L and SUZ12 proteins which are components of MLL and PRC2/EED-EZH1 complexes which regulate histone H3 methylation (*Patel et al., 2009*; *Pasini et al., 2004*); SETD1A, HCFC1, HDAC2, YY1 and RCOR1, which are components of H3K4 demethylase complexes and co-repressors; and UHRF1, which binds to H3K9me3-modified chromatin and is involved in both maintaining DNA methylation and heterochromatin formation (*Bostick et al., 2007*; *Guetg et al., 2010*; *Rottach et al., 2010*). Ki-67 also interacts with TIP5, the major component of the nucleolar remodelling complex (NoRC) which is required to establish and maintain perinucleolar heterochromatic rDNA, as well as NoRC-interacting proteins TTF1 and DNM3.

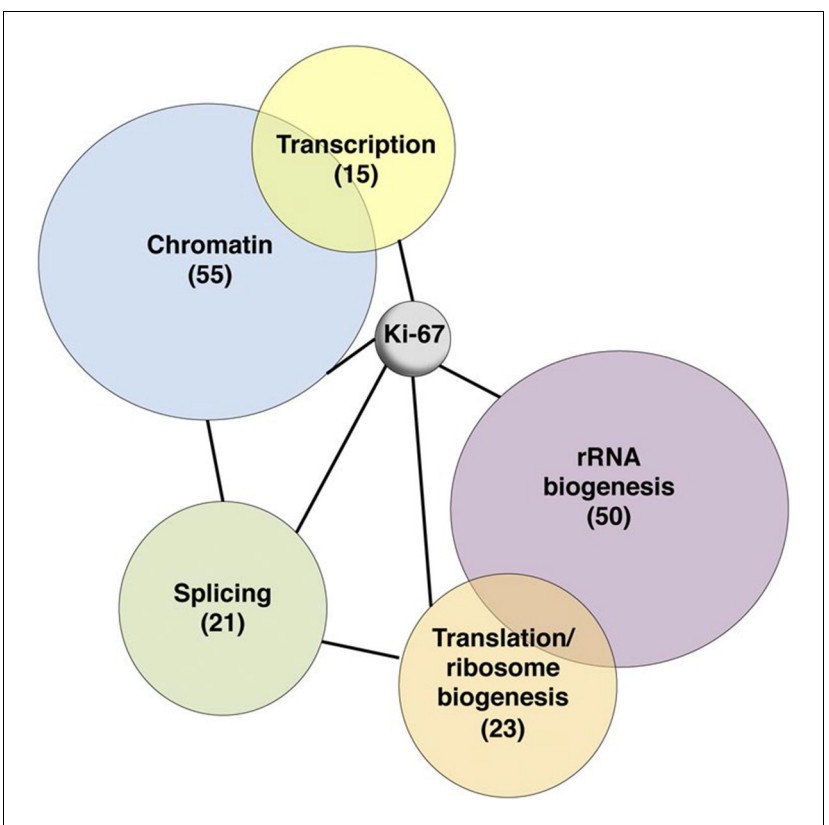

**Figure 6.** Ki-67 interacts with proteins involved in nucleolar processes and chromatin. Simplified STRING analysis reveals network interactions between proteins associating with Ki-67. The full network is shown in *Figure 6—figure supplement 2*; data is provided in *Figure 6—source data 1*.

The following source data and figure supplements are available for figure 6:

**Source data 1.** Ki-67 interacting proteome.

**Figure supplement 1.** The Ki-67 interactome.

**Figure supplement 2.** Ki-67 interacts with proteins involved in nucleolar processes and chromatin.

## Ki-67 is required for perichromosomal region formation during nucleologenesis

We first focused on a possible role of Ki-67 in ribosome biogenesis, a process linked to nucleolar assembly and structure, and which is required for cell proliferation (*Hernandez-Verdun et al., 2010*). One of the candidate Ki-67 interactors involved in rRNA biogenesis was the Pescadillo homologue PES1, which participates in pre-rRNA processing and is localised in the nucleolar granular components (GC) (*Rohrmoser et al., 2007*; *Tafforeau et al., 2013*). During interphase, we found that Ki-67 localised at the cortical periphery of the GC, visualised using PES1. Ki-67 formed a boundary between the perinucleolar heterochromatin (clearly visible as a 'ring' in the DAPI staining) and the GC (*Figure 7A*, *Figure 7—figure supplement 1*). Whereas nucleolar disruption using Actinomycin D or the CDK inhibitors DRB or Roscovitine caused nuclear relocalisation of Ki-67 and GC proteins (*Figure 7—figure supplement 2*), depletion of Ki-67 did not affect the gross structure of the nucleolus, as determined by PES1 staining (*Figure 7A*, *Figure 7—figure supplement 3*).

During mitosis, the nucleolus undergoes a dramatic cycle of disassembly and reassembly (*Hernandez-Verdun et al., 2010*). Briefly, soon after the onset of mitosis, when transcription is shut down, the nucleolus is rapidly disassembled; it then slowly reforms through the formation of intermediary organelles that undergo consecutive transformations, identifying three distinct organelle stages, and the process is complete by telophase. The first of these three intermediary organelles is a sheath of nucleolar proteins that forms around the surface of the mitotic chromosomes, the so-called 'perichromosomal region' or PR. To date, not much is known about the *trans*-acting factors involved in PR formation. Remarkably, we found that Ki-67 depletion totally disrupted PR formation and PES1 no longer associated with the chromosome surface (*Figure 7B*). A similar finding, using other nucleolar proteins than PES1 as PR markers, was recently reported (*Booth et al., 2014*).

Having established that Ki-67 controls nucleolar assembly during mitosis, we wondered whether Ki-67 is required for pre-rRNA processing. We found that Ki-67 knocked down U2OS cells, that have essentially undetectable Ki-67, could still incorporate normal levels of 5-ethynyl uridine (EU) in nucleolar RNA. This suggests that rRNA transcription is not altered (*Figure 8—figure supplement 1*). We next looked at pre-rRNA processing pathways by Northern blotting of precursor rRNAs or intermediates (*Figure 8—figure supplement 2*). This showed that silencing Ki-67 expression by shRNA or siRNA had no significant effect on pre-rRNA processing in four different cancer cell lines (*Figure 8A*). We did, however, notice a marginal but reproducible increase in the level of the 47S precursor rRNA, indicating a mild delay in the early nucleolar pre-rRNA cleavage steps (*Figure 8B*). The tumour suppressor TP53 (p53) is a sensor of nucleolar stress resulting from defective ribosome biogenesis, and it represses ribosomal gene transcription (*Bursac et al., 2014*). Impairment of early pre-rRNA cleavage steps upon depletion of Ki-67 was independent of p53 (*Figure 8B*). Taken together, these results demonstrate that while Ki-67 is dispensable for efficient pre-rRNA processing, it is essential for the formation of the perichromosomal layer during nucleologenesis.

## Depletion of Ki-67 affects gene transcription

These results showed that in spite of its role in early steps of nucleologenesis, Ki-67 is not essential for expression of rRNA genes. We next asked whether it is involved in control of mRNA expression. We performed genome-wide transcriptome analysis from U2OS and HeLa cells expressing non-silencing control or Ki-67 shRNA, using Agilent gene-microarrays. Ki-67 knockdown led to downregulation (corrected p value <0.02, Fold-change >1.5) of over 200 genes (*Figure 8C*). Expression of cell cycle regulatory genes was not affected. Additionally, Ki-67 silencing caused upregulation (corrected p value <0.02, Fold-change >1.5) of a wide variety of genes involved in neural, testis and cardiovascular system development and differentiation (*Figure 8C*). These were strikingly enriched in genes encoding zinc-finger proteins and olfactory receptors, two gene families that are highly enriched in nucleolar associated-domains (NADs) of perinucleolar heterochromatin (PNHC) (*Németh et al., 2010*). This suggested that effects of Ki-67 downregulation on gene expression might be due to an altered chromatin state, in particular of PNHC.

## Ki-67 promotes heterochromatin organisation

In support of a potential role for Ki-67 in heterochromatin organisation, we found that Ki-67 knockdown in HeLa and U2OS cells caused a marked reduction in perinucleolar DAPI staining (*Figure 7A*,

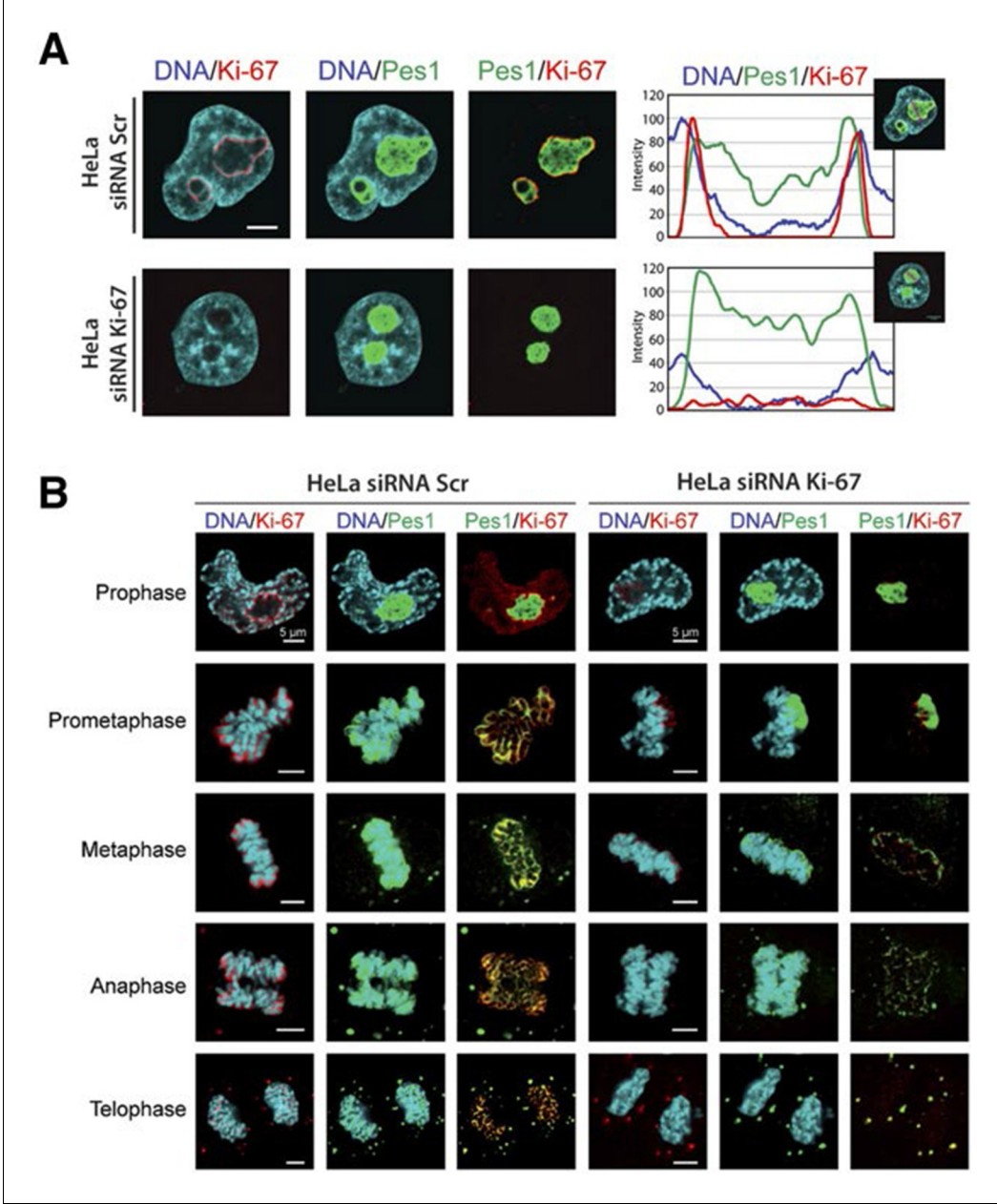

**Figure 7.** Ki-67 localises PES1 to mitotic chromosomes. (**A**) Analysis of the interphase localisation of PES1 and Ki-67 proteins by immunofluorescence in HeLa cells 72 hr after transfection with control siRNA (scramble; Scr) or Ki-67 RNAi. Right, line scans showing the distribution of fluorescence signals within indicated nucleoli (dashed line). Images were captured in confocal mode with a spinning-disk microscope. Bar, 5 μm. (**B**) Analysis of the mitotic localisation of PES1 and Ki-67 proteins by immunofluorescence in HeLa cells 72 hr after transfection with control siRNA (scramble; Scr) or Ki-67 RNAi. Bar, 5 μm.

The following figure supplements are available for figure 7:

**Figure supplement 1.** Ki-67 is a nucleolar protein localizing in the cortical side of the GC.

**Figure supplement 2.** Ki-67 follows GC components upon drug-induced nucleolar disruption.

**Figure supplement 3.** Depletion of Ki-67 does not affect overall nucleolar structure.

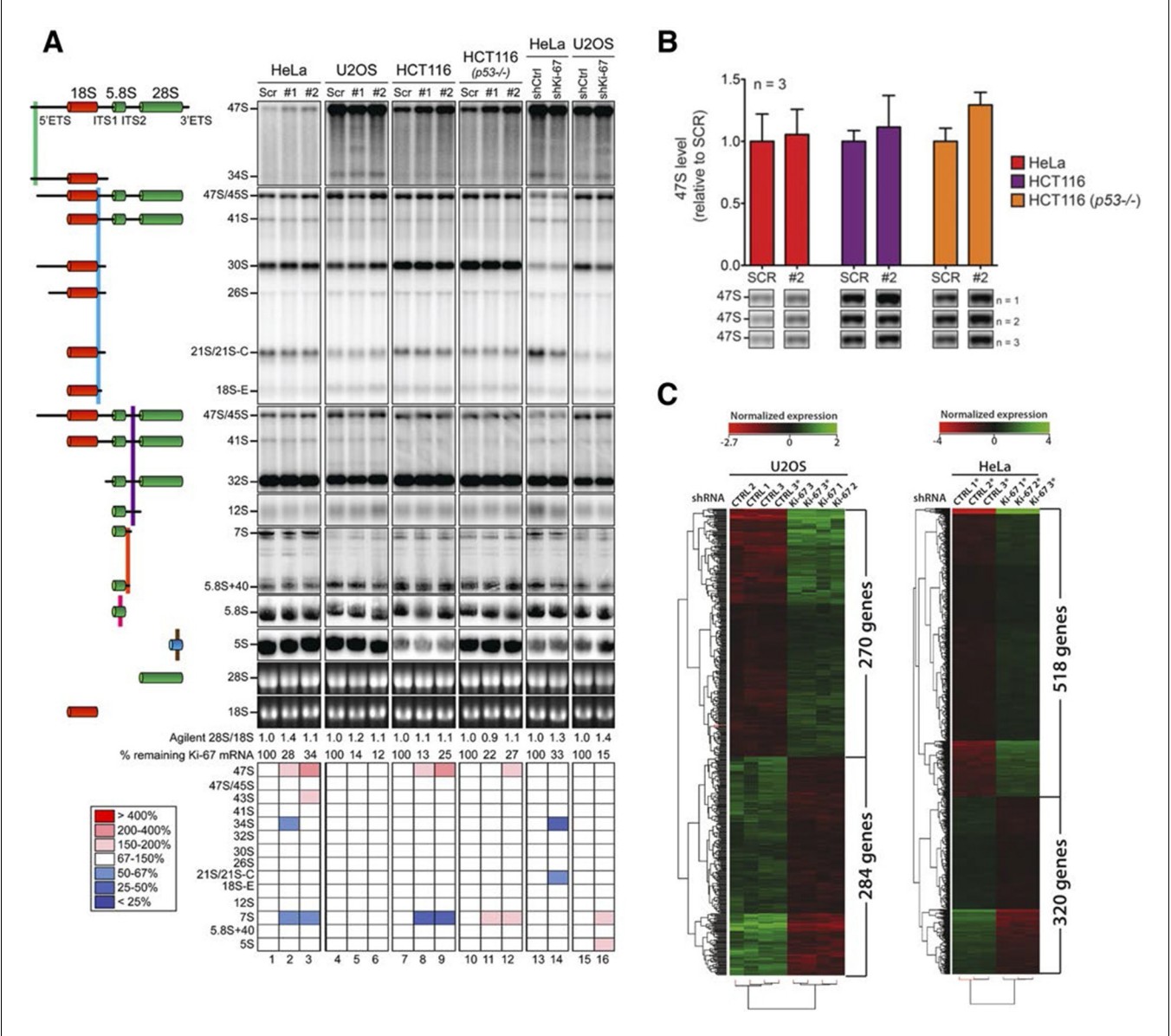

**Figure 8.** Ki-67 is not required for rRNA biogenesis but controls gene transcription. (**A**) Northern-blot analysis of total RNA extracted from HeLa and U2OS cells constitutively expressing shRNA against Ki-67; and HeLa, U2OS, HCT-116 and HCT-116 TP53 (-/-) depleted of Ki-67 by siRNA for 72 hr in two biological replicates (#1 and #2) or with scrambled siRNA control (Scr). Pre-rRNA intermediates were analysed by probing with different primers located in the different spacers of the 47S sequence (5'ETS-green; ITS1-blue; ITS2-purple). (**B**) Quantification of 47S rRNA precursor in HeLa, HCT-116 and HCT-116 TP53 (-/-) depleted of Ki-67 by siRNA for 72 hr in three biological replicates (n=1–3). (**C**) U2OS cells (left) or HeLa cells (right) show transcriptome profile differences (fold change >1.5; corrected p-value <0.02) between asynchronous cells constitutively expressing control (CTRL) or Ki-67 shRNA. Heat-maps present the expression levels of differentially expressed genes between biological replicates (1,2,3) and technical replicates (3, 3*). Data is provided in *Figure 8—source data 1* and *2*.

The following source data and figure supplements are available for figure 8:

**Source data 1.** Ki-67-dependent transcriptome in U2OS cells.

**Source data 2.** Ki-67-dependent transcriptome in HeLa cells.

**Figure supplement 1.** Ki-67 depletion does not hinder rRNA transcription.

**Figure supplement 2.** Human pre-rRNA processing pathway involves two major pathways.

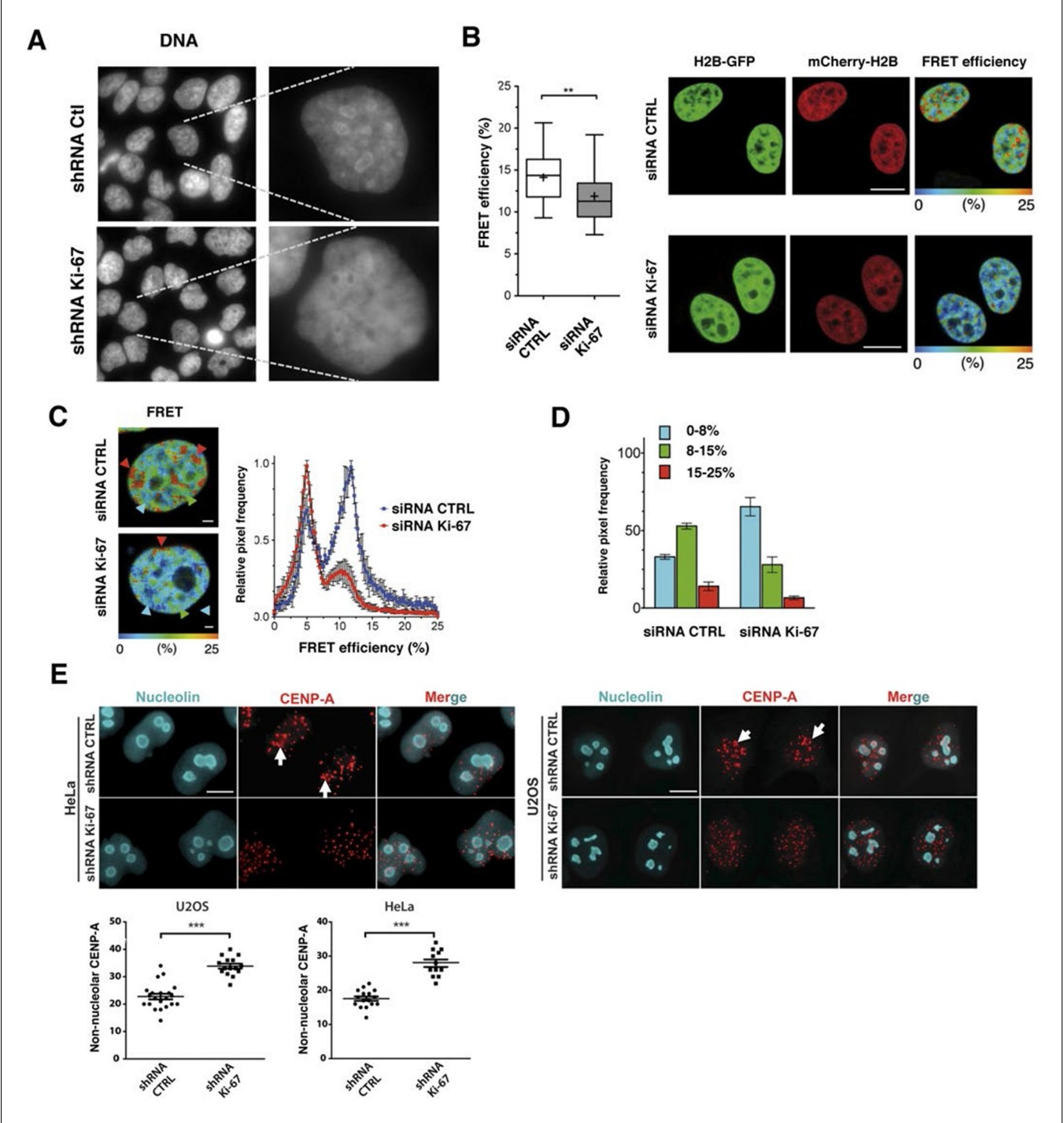

**Figure 9.** Ki-67 promotes heterochromatin interactions. (**A**) DAPI staining in control and stable Ki-67-knockdown U2OS cells. (**B**) HeLa cells stably expressing GFP-H2B and mCherry-H2B, depleted using Ki-67 or non-targeting (CTRL) siRNA. Left, FRET efficiency (cross shows mean value) ** Different, p<0.01. FRET efficiency and spatial distribution shown by a pseudocolour scale of FRET (%) values from 0 to 25%. Bars, 10 μm. (**C**) Left, representative HeLa^H2B-2FP nuclei showing spatial distribution of FRET efficiency. Arrowheads show different chromatin compaction states (high FRET, red; intermediate, green; low, blue), Bars, 2 μm. Right, mean FRET distribution curves from siRNA control (blue curve, n=8) and siRNA Ki-67 (red curve, n=11) nuclei. (**D**) Relative fraction of the three FRET efficiency populations (blue (low), FRET efficiency ≤ 8%; green (medium), 8–15%; and red (high), 15–25%) in siRNA control and siRNA Ki-67 nuclei. Error bars indicate SD. (**E**) Immunofluorescence of CENP-A localisation in control and stable Ki-67 knockdown HeLa (left) and U2OS (right) cells. Nucleolar localisation (white arrows). Nucleolin was used as nucleolar marker. Bar, 10 μm. Below: quantification in different cells of numbers of CENP-A spots not associated with the nucleolus.

*Figure 9 continued on next page*

*Figure 9 continued*

The following figure supplement is available for figure 9:

**Figure supplement 1.** Knockdown of Ki-67 in H2B FRET cell line.

*9A*). We thus hypothesised that Ki-67 might be required for heterochromatin compaction. To directly assess chromatin compaction in living cells, we used a Förster Resonance Energy Transfer-Fluorescence Lifetime Imaging Microscopy (FRET-FLIM)-based assay. In this system, HeLa cells stably co-express versions of histone H2B labelled with eGFP and mCherry. Inter-nucleosomal interactions between H2B-eGFP and H2B-mCherry generates FRET, whose efficiency depends on the distance between nucleosomes (*Llères et al., 2009*). We depleted Ki-67 by siRNA and studied the effects on FRET efficiency in interphase cells (*Figure 9—figure supplement 1*). As expected (*Llères et al., 2009*), a heterogeneous FRET efficiency map was observed throughout control siRNA nuclei, reflecting different chromatin compaction states (*Figure 9B*). Upon Ki-67 depletion, the mean FRET percentage decreased, reflecting a reduction in total chromatin compaction (*Figure 9B*). The highest FRET populations, including heterochromatin regions at the nuclear periphery and around nucleoli, were largely eliminated upon Ki-67 depletion, with a few remaining condensed foci predominantly located at the nuclear periphery (*Figure 9C,D*). In contrast, a population of less compact chromatin increased.

Reduced compaction of heterochromatin implies disruption of short-range interactions of chromatin. To determine whether Ki-67 knockdown also affects long range interactions, we assessed interactions between perinucleolar and pericentromeric heterochromatin. We stained for the centromeric histone variant CENP-A to determine the localisation of centromeric DNA. In HeLa and U2OS cells, CENP-A showed a non-random nuclear localisation and clustered around nucleoli (*Figure 9E*, arrowheads), confirming that CENP-A can be used as a surrogate marker for adjacent pericentromeric DNA. Consistent with our hypothesis, this interaction was disrupted upon Ki-67 knockdown, and the CENP-A signal was no longer grouped around nucleoli but dispersed throughout the nucleus (*Figure 9E*). These results imply that Ki-67 mediates interaction between different regions in the genome that are normally packaged into heterochromatin, and could potentially maintain silencing of genes by recruiting them to constitutive heterochromatin.

Constitutive heterochromatin, including PNHC, is characterised by histone post-translational modifications H3K9me3 and H4K20me3. We asked whether they were affected by downregulation of Ki-67. Stable shRNA-mediated Ki-67 knockdown in HeLa, U2OS and inducible knockdown in BJ-hTERT fibroblasts caused a visible reduction in nucleolar staining of H3K9me3 and H4K20me3 (*Figure 10—figure supplements 1–3*). This mark was relocalised either to foci in proximity to the nucleolus or a punctate pattern dispersed throughout the nucleus. In mouse cells, where pericentromeric heterochromatin is prominent, H3K9me3 staining colocalised with DAPI-dense chromocentres, but TALEN-ablation of *Mki67* resulted in general nuclear punctate H3K9me3 that was excluded from nucleoli (*Figure 10A,B*). Western blotting revealed that Ki-67 depletion did not affect the overall levels of these chromatin modifications (*Figure 10—figure supplement 4*). We also analysed the localisation of heterochromatin protein 1 (HP1), which binds to chromatin containing H3K9me3. Immunofluoresence showed that, surprisingly, despite the loss of the intense H3K9me3 staining regions in the *Mki67* mutant cells, all three HP1 isoforms maintained their localisation at DAPI-dense regions (*Figure 10—figure supplement 5*). Next, we determined whether heterochromatic histone marks were reduced on specific DNA sequences, or whether they were retained but the sequences themselves were delocalised. To do this we examined co-localisation of H3K9me3 with mouse major satellite DNA, by combining immunofluorescence with fluorescent in situ hybridisation (FISH). In control cells, major satellite DNA, DAPI-dense regions and H3K9me3 largely colocalised (*Figure 10C*). In cells lacking Ki-67, major satellite DNA was still present at regions of compacted DNA, despite the loss of H3K9me3 at these regions (*Figure 10C*). Taken together, these results suggest that Ki-67 is required for maintaining heterochromatic histone marks at genomic regions that are organised into heterochromatin.

These results suggest that Ki-67 is required for heterochromatin organisation. To see whether Ki-67 is sufficient to promote heterochromatin formation, we cloned a full-length cDNA encoding human Ki-67, that we fused to the eGFP gene, and transfected it into U2OS cells. There was a strong

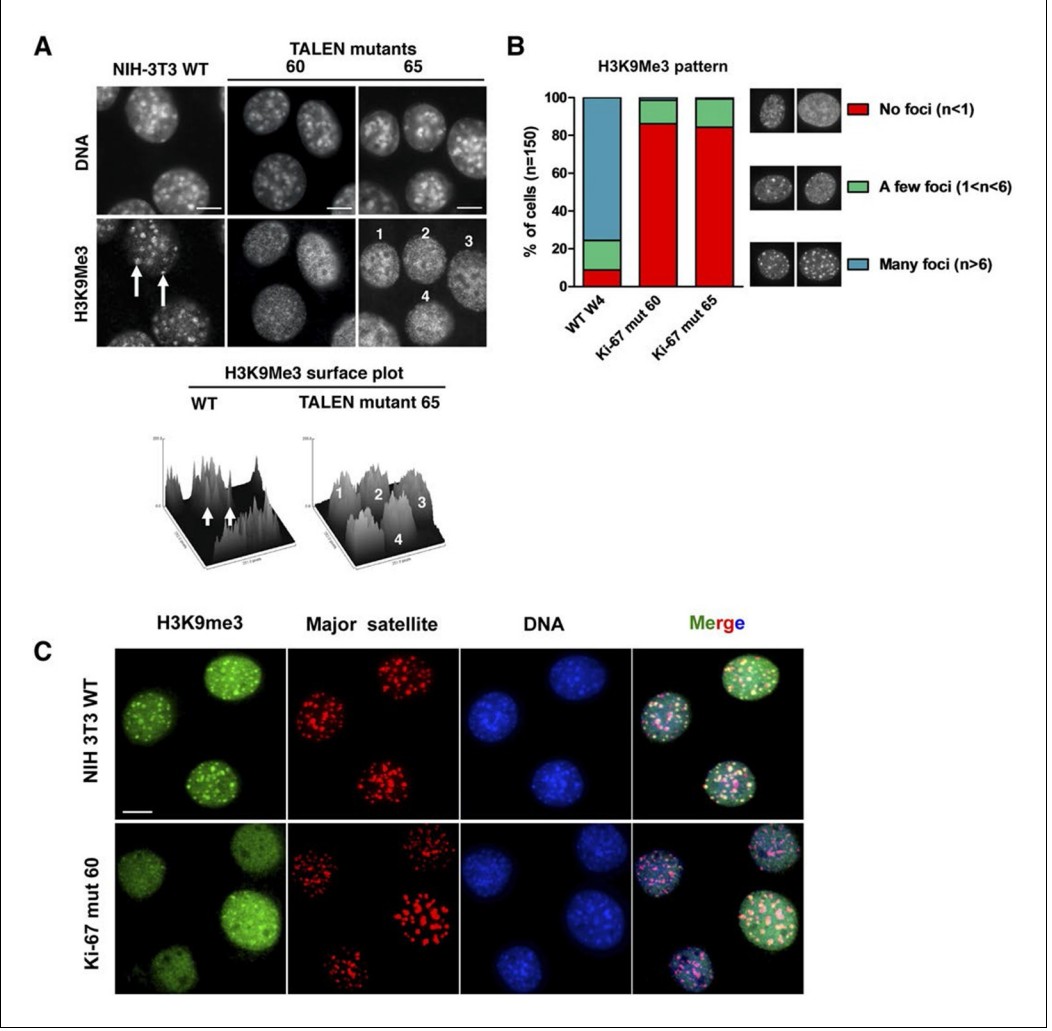

**Figure 10.** Ki-67 controls heterochromatin organisation. (**A**) Top, immunofluorescence analysis of H3K9me3 in mouse NIH-3T3 WT and *Mki67* TALEN mutant clones 60 and 65. Bars, 5 μm. Below: graphs showing quantification of pixel intensity scans for H3K9me3. (**B**) Quantification of H3K9Me3 patterns in WT and Ki-67 mutant clones. (**C**) Immunofluorescence of H3K9Me3, FISH of major satellite DNA and DAPI staining in WT W4 and Ki-67 mutant clone 60. Bar, 10 μm.

The following figure supplements are available for figure 10:

**Figure supplement 1.** Heterochromatic histone mark localisation requires Ki-67.

**Figure supplement 2.** Heterochromatic histone mark localisation requires Ki-67.

**Figure supplement 3.** Heterochromatic histone mark localisation requires Ki-67.

**Figure supplement 4.** Overall heterochromatic histone mark levels do not change upon Ki-67 knockdown.

**Figure supplement 5.** HP1 localises normally to chromocentres in Ki-67 mutant cells.

correlation between cells with higher levels of exogenous Ki-67 and appearance of DAPI-dense foci resembling ectopic heterochromatin, marked by H3K9me3 and HP1 (*Figure 11A*). Cells showing this phenotype were negative for cyclin A staining, suggesting that they were unable to enter S-phase, whereas lower expression did not prevent cyclin A accumulation (*Figure 11B*). They were also negative for histone H3 Ser-10 phosphorylation, a mitotic marker (*Figure 11—figure supplement 1*). If

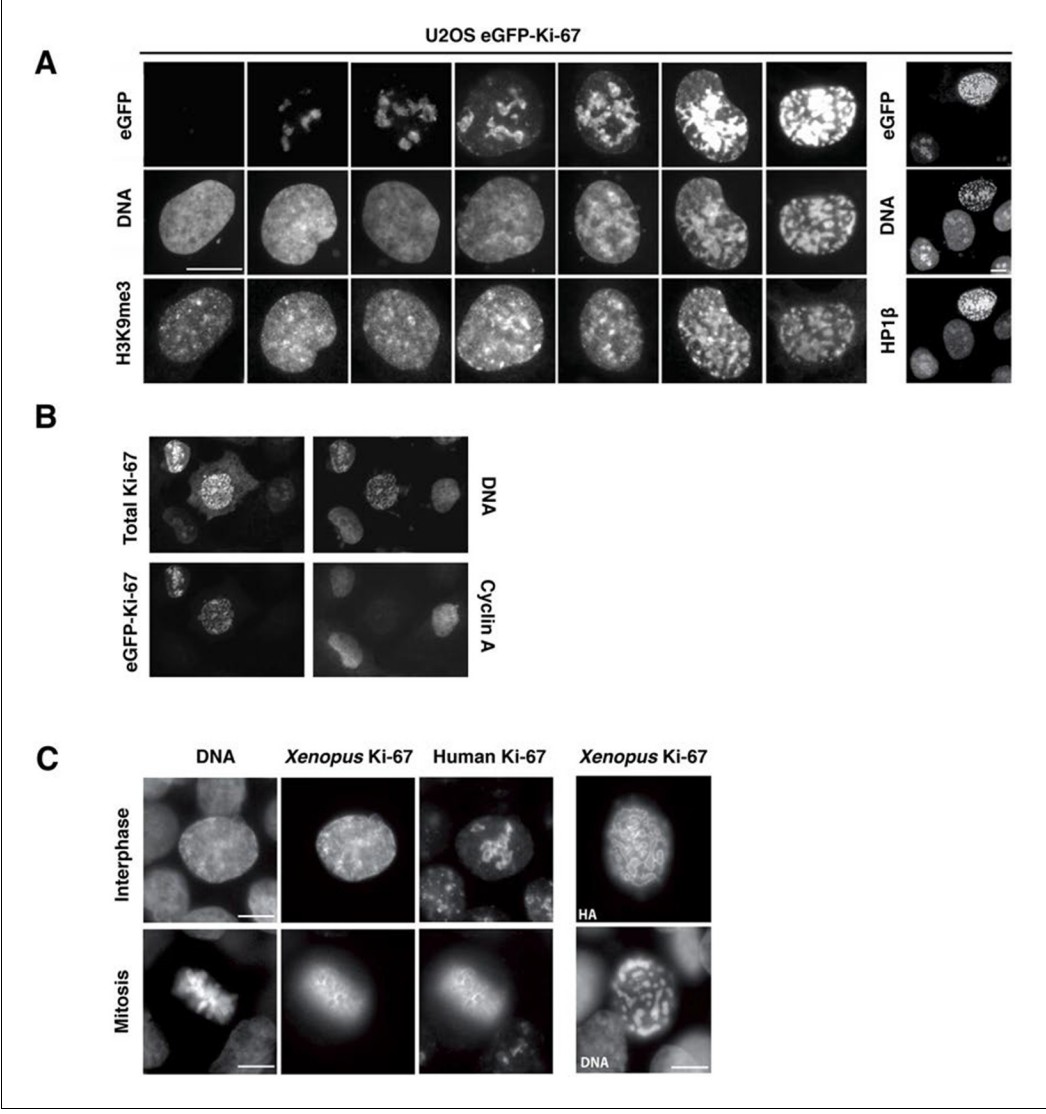

**Figure 11.** Overexpression of Ki-67 induces ectopic heterochromatin. (**A**) Overexpression of full length Ki-67 N-terminal fusion with eGFP in U2OS cells induces ectopic heterochromatin, as visualised by DAPI staining (middle) and immunofluorescence of H3K9Me3 (left) or HP1β (right). Eight representative cells that have different levels of Ki-67 expression, as determined by eGFP fluorescence intensity, are shown. Bar, 10 μm. (**B**) U2OS cells expressing high levels of exogenous eGFP-Ki-67 and showing ectopic chromatin condensation are negative for cyclin A staining by immunofluorescence. (**C**) Left: Immunofluorescence analysis of the localisation of endogenous human and ectopically expressed *Xenopus* Ki-67 in U2OS cells, showing colocalisation in metaphase at the perichromosomal region. Right: DNA condensation caused by high overexpression of *Xenopus* Ki-67 in U2OS cells. Bars, 10 μm.

The following figure supplements are available for figure 11:

**Figure supplement 1.** Overexpression of Ki-67 induces ectopic heterochromatin.

**Figure supplement 2.** A Xenopus Ki-67 homologue (**A**) Chromatin proteomics in replicating *Xenopus* egg extracts.

controlling heterochromatin organisation is a major function of Ki-67, it is likely to be conserved in more distantly related Ki-67 homologues. In a proteomics-based screen for proteins associated with replicating chromatin in egg extracts, we identified a putative *Xenopus* Ki-67 homologue (*Figure 11—figure supplement 2*). To assess whether *Xenopus* and human Ki-67 are functionally conserved, we cloned and HA-tagged full-length *Xenopus* Ki-67 and expressed it in U2OS cells. Whereas in interphase, exogenous *Xenopus* Ki-67 is present ubiquitously on chromatin, it colocalises

with endogenous Ki-67 in mitosis at the perichromosomal region (*Figure 11C*). Overexpression of *Xenopus* Ki-67, like human Ki-67, caused extreme chromatin compaction (*Figure 11C*). We conclude that controlling heterochromatin is a conserved essential function of Ki-67.

## Discussion

Ki-67 has long been assumed to be essential for cell proliferation (*Schluter et al., 1993*; *Kausch et al., 2003*; *Starborg et al., 1996*; *Rahmanzadeh et al., 2007*; *Zheng et al., 2006*; *2009*). Using various genetic and knockdown approaches, we found no evidence for this in any cell type we tested: HeLa, U2OS, BJ-hTERT and NIH-3T3 fibroblasts. Our data show that Ki-67 expression can be uncoupled from cell proliferation in both directions. Indeed, not only can cells lacking Ki-67 proliferate efficiently, conversely, interfering with Ki-67 downregulation by disrupting the *Cdh1* gene did not prevent cell cycle exit in vivo. It remains possible that certain cell lines are more sensitive to inhibition of Ki-67 expression, eg cancer cell lines of bladder (*Kausch et al., 2003*) or renal (*Zheng et al., 2006*; *2009*) origins. Alternatively, off-target effects of previous antisense or RNAi approaches might have contributed to the cell proliferation defects observed in previous studies, as none of them employed restoration controls using silencing-insensitive constructs. Such rescue experiments are virtually unfeasible given the large size of Ki-67, the targeting of silencing oligonucleotides to the repeated domains, and the fact that Ki-67 overexpression induces ectopic heterochromatin. Several other studies used different approaches. In one, microinjection of an anti-Ki-67 antibody in 3T3 cells did not cause an abolition of cell division (*Starborg et al., 1996*). Instead, there was a modest reduction, from 80% to 64%, of dividing cells over a 36-hr period. Another study used chromophore-mediated light inactivation of Ki-67 after injecting chromophore-labelled Ki-67 antibodies, and found an inhibition of ribosomal RNA synthesis (*Rahmanzadeh et al., 2007*). However, such an approach might cause non-specific collateral damage to nucleolar processes where Ki-67 is localised.

We found that rather than promoting cell proliferation, the role of Ki-67 is to organise heterochromatin. We showed that Ki-67 is required for the maintenance of a high level of compaction typical of heterochromatin, and mediates long-range interactions between different regions of the genome that are packaged into heterochromatin. We speculate that heterochromatin compaction relies on local interactions that depend on Ki-67. Cells lacking Ki-67 show altered gene expression profiles upon long-term Ki-67 silencing, with a striking correlation between upregulation of genes that normally are physically associated with perinucleolar heterochromatin (*Németh et al., 2010*). To determine possible mechanisms of action of Ki-67, we comprehensively identified its interacting partners. We thus found at least seventeen proteins that are involved in histone methylation complexes or are interactors of methylated chromatin required for heterochromatin maintenance. This suggests that Ki67 might target these proteins to their genomic sites to promote heterochromatin formation. Consistent with this hypothesis, Ki-67 downregulation led to reduction of H3K9me3 and H4K20me3 at heterochromatin, while Ki-67 overexpression caused appearance of ectopic heterochromatic foci enriched in these methylation marks. Unexpectedly, Ki-67 downregulation did not prevent association of HP1 isoforms with heterochromatin. Possibly, low levels of H3K9me3 or H4K20me3 persist and are sufficient for recruitment of HP1, or alternative mechanisms exist to localise HP1 to chromatin. The former hypothesis would be consistent with the observation that loss of the Suv39H methyltransferases that are responsible for H3K9me3 and H4K20me3 abrogates HP1 recruitment to heterochromatin in mice, whose late embryonic growth and survival is impaired (*Peters et al., 2001*). Nevertheless, evidence for the latter possibility has been provided by a study in *C.elegans*, in which genome-wide distribution of HP1 binding, as assessed by ChIP-seq, was conserved in animals lacking H3K9 di- and trimethylation (*Garrigues et al., 2015*).

Given the requirement for Ki-67 expression in organising heterochromatin, it is perhaps surprising that mouse development is not affected by Ki-67 downregulation. This once again highlights the robustness of biological systems. However, to determine whether mouse development can occur normally in the complete absence of Ki-67 will require a gene deletion rather than a gene disruption mediated by genome-editing, as we found that even deletion of the translation initiation ATG using TALENs did not completely abolish Ki-67 expression. The eight subsequent ATG codons are out of frame, and the next in-frame ATG is 433 bp downstream. Translation from any frameshifted ATG will lead to a premature stop codon within 65 nucleotides. Although NMD of the mRNA did not occur,

translation was strongly reduced. This is probably due to the presence of many out-of-frame ATG codons before the next in-frame ATG codon, as well as the distance from the 5' end of the mRNA. It is likely that the residual Ki-67 in proliferating cells occurred from the next in-frame ATG, thus eliminating the most highly conserved domain of Ki-67, the Forkhead-associated (FHA) domain. However, the unexpected translation in the mutants suggests that care should be taken to examine possible low level expression of proteins after mutating start sites using genome editing approaches.

Since Ki-67 is degraded upon cell cycle exit, we speculate that this may alter chromatin structure. For example, Ki-67 degradation might be involved in heterochromatin rearrangements observed during senescence onset. Facultative heterochromatic foci, that characterise some senescent cells (*Narita et al., 2003*; *2006*) are not a consistent feature of senescence in all cell types. In contrast, large-scale satellite heterochromatin decondensation is an early step in senescence in all cells studied and it precedes loss of H3K9me3 (*Swanson et al., 2013*). As Ki-67 is required for heterochromatin compaction, its degradation may be involved in the heterochromatin decompaction occurring upon senescence onset. Heterochromatin reorganisation caused by Ki-67 downregulation does not interfere with cell cycle progression or cell proliferation, but likely contributes to remodelling of gene expression. Heterochromatin is also less compact in highly proliferative pluripotent stem cells, suggesting that heterochromatin organisation is critical for determining transcriptional responses (*Fussner et al., 2011*). Ki-67 overexpression, which led to pronounced chromatin condensation, appeared to arrest the cell cycle in G1, implying that controlled Ki-67 degradation is required to allow unperturbed progression through the cell cycle.

The nucleolus is a potent cancer biomarker (*Derenzini et al., 2009*), and a recently demonstrated target in cancer therapy. Inhibitors of nucleolar functions have indeed been shown to selectively kill cancer cells, leaving non-cancerous cells intact (*Bywater et al., 2012*; *Peltonen et al., 2014*). It is therefore critical to understand how the nucleolus forms during mitosis. An important step in nucleolar assembly is the formation of a sheath of nucleolar proteins around the chromosome surface on the metaphase plate. This so-called perichromosomal layer has been suggested to play roles in chromosome protection, in the faithful partitioning of nucleolar proteins between daughter cells, and in the segregation of opportunistic passenger proteins. Whether the PR performs any of these functions or has other, unidentified, roles will be a promising field for future studies. In this study, we have identified Ki-67 as one of the first *trans*-acting factors involved in PR formation during nucleologenesis, corroborating a recent report (*Booth et al., 2014*).

In conclusion, our data reveal a novel concept whereby heterochromatin organisation is linked to cell proliferation by Ki-67. As heterochromatin organisation is often compromised in cancer cells (*Carone and Lawrence, 2013*) and Ki-67 expression is widely used in clinical assessments in cancer, these data provide a rationale for further investigation of the functional consequences of Ki-67 expression in tumour samples. Importantly, our data suggest that Ki-67 is likely to modulate transcription in cancer cells.

## Materials and methods

### Ethics
All animal experiments were performed in accordance with international ethics standards and were subjected to approval by the Animal Experimentation Ethics Committee of Languedoc Roussillon.

### Cell lines
Normal human diploid foreskin fibroblasts (HDF) were provided by Jacques Piette (CRBM, Montpellier), the *hTERT*-immortalized foreskin fibroblast cell line (BJ hTERT) was provided by Jean Marc Lemaitre (IRB, Montpellier). U2OS, HeLa, NIH3T3 mouse fibroblasts were obtained from the American Type Culture Collection. They were not authenticated but were mycoplasma-free (tested weekly). U2OS, HeLa and NIH 3T3 were grown in Dulbecco modified Eagle medium (DMEM - high glucose, pyruvate, GlutaMAX – LifeTechnologies, ThermoFisher Scientific, Paris, France) supplemented with 10% foetal bovine serum (Sigma-Aldrich, Lyon, France or HyClone, GE Healthcare, Paris, France). BJ hTERT were grown in DMEM supplemented with 10% foetal calf serum (Sigma-Aldrich) and 2 mM L-glutamine. Apart from murine embryo fibroblasts (MEFs), cells were grown under standard conditions at 37°C in a humidified incubator containing 5% $CO_2$. MEFs were grown

in DMEM supplemented with 10% fetal bovine serum and 1% Penicillin/Streptomycin at 37°C in an incubator containing 3% $O_2$ and 5% $CO_2$.

## Cell synchronisation

### G0 block

HDF, BJ hTERT and NIH-3T3 cell lines at 20% confluency were washed once with PBS and incubated with medium supplemented with 0.1% FBS for 72 hr. Cells were stimulated to enter the cell cycle by adding fresh medium with 10% FBS. Onset of S phase was observed at 16 hr after restimulation by EdU incorporation assay.

## Lentiviral infection

The lentiviral constitutive and inducible knockdown systems were packaged into non-replicating lentivirus HIV-1 using II generation packaging system – psPAX by PVM platform (IGF, Montpelier). Immortalized BJ-hTERT and U2OS cells were infected at MOI 10 with lentivirus armed pTRIPZ GAPDH positive CTRL and Ki-67 inducible shRNA, and immortalized BJ hTERT, U2OS and HeLa were infected at MOI 10 with lentivirus armed pGIPZ shRNA non-silencing and Ki-67 constitutive system. Lentiviral transduction was performed according to the manufacturer's protocol (Thermo Scientific). 2 days after infection cells were selected with 10 µg/ml puromycin for 4 days. Cells were treated with progressively increased puromycin concentrations up to 60 µg/ml, to select the most highly transduced population.

## shRNAs

The lentiviral constitutive knockdown vectors containing shRNAs were purchased from ThermoFisher Scientific.

1. pGIPZ shRNAmir Ki-67 (clone ID: V2LHS-151787)
AGGCTACAAACTCGTAAGGAAATAGTGAAGCCACAGATGTATTTCCTTACGAGTTTGTAGCCG
2. pGIPZ shRNAmir CTRL non-targeting (RHS4346)
ACCTCCACCCTCACTCTGCCATTAGTGAAGCCACAGATGTAATGGCAGAGTGAGGGTGGAGGG

The lentiviral doxycycline-inducible knockdown positive control vector containing shRNA GAPDH was purchased from Thermo Scientific.

3. pTRIPZ shRNAmir GAPDH
CCCTCATTTCCTGGTATGACAATAGTGAAGCCACAGATGTATTGTCATACCAGGAAATGAGGT

pTRIPZ shRNAmir Ki-67 vector was obtained by sub-cloning to replace the shRNAmir GAPDH in pTripZ with the shRNAmir sequence from pGIPZ shRNAmir Ki-67.

## Induction of shRNA expression in cells transduced with inducible pTripZ shRNA

To induce expression of shRNA, cells were treated with 2 µg/ml doxycycline hyclate (Sigma-Aldrich) for minimum 24 hr and then during the period of designed experiments. Downregulation of mRNA of shRNA target genes was analysed 24 hr post induction.

## siRNA transfection

The SMARTpool: ON-TARGETplus siRNAs were purchased from GE Dharmacon (Lafayette, CO, USA). Cells were transfected with SMARTpool: ON-TARGETplus siRNA non-targeting (D-001810-10), *MKI67* (L-003280-00) and *FZR1* (L-015377-00) at 100 nM by Calcium Phosphate transfection method.

## Calcium phosphate transfection

### Materials

2.5 M $CaCl_2$, 2x HBSS buffer (50 mM HEPES, 280 mM NaCl, 1.5 mM $Na_2HPO_4$, 10 mM KCl; pH 7,04), sterile $H_2O$.

Cells were plated at density of $1.5x10^4/cm^2$ in the afternoon the day before transfection. 30 min before transfection, growing medium with antibiotics were exchanged for 2 ml of growing medium without antibiotics, if cells were plated on 21 $cm^2$. Calcium phosphate–DNA coprecipitate were

prepared by pipetting 112.5 µL sterile $H_2O$, 12.5 µL of 2.5 M $CaCl_2$ and 2 µL of 100 µM siRNA (final concentration - 100 nM in medium above cells), without vortexing. $CaCl_2$-siRNA solution were combined with equal volume of 2xHBSS buffer. Coprecipitates were incubated at room temperature for 5 min, mixed by pipetting, added drop by drop into medium above cells and distributed by moving back and forward.

## Cell extracts and western-blotting

Frozen pellets (harvested by trypsinization, washed with cold PBS) were lysed directly in Laemmli buffer at 95°C (without β-mercaptoethanol and bromophenol blue) and sonicated in a chilled bath for 10 min in 30 s/30 s ON/OFF intervals. Protein concentrations were determined by BCA protein assay (ThermoFisher). Equivalently loaded proteins were separated by SDS-polyacrylamide gel electrophoresis (SDS-PAGE) (usually on 12 cm x 14.5 cm 7.5% and 12.5% gels) at 35 mA in TGS buffer (25 mM Tris, 200 mM glycine, 0.1% SDS). The proteins were then transferred to Immobilon membranes (Milipore) at 1.15 mA/cm$^2$ for 120 min with a semidry blotting apparatus containing transfer buffer (25 mM Tris, 200 mM Glycine, 0.2% SDS, 20% EtOH). Membranes were blocked in TBST pH 7.6 (20 mM Tris, 140 mM NaCl, 0.1% Tween-20) containing non-fat dry milk (5%), incubated with antibody for 2 hr at RT with agitation in TBST containing non-fat dry milk (1,25%), washed several times with TBST for a total of 45 min, incubated with secondary antibody at 1/5000 dilution in TBST containing non-fat dry milk (1.25%) for 1 hr at RT with agitation and washed several times for 1 hr in TBST. Secondary antibodies were either goat antibodies to mouse IgG-HRP (immunoglobulin G – horseradish peroxidase) (DACO) or donkey antibodies to rabbit IgG-HRP (immunoglobulin G – horseradish peroxidase) (GE Healthcare). The detection system used was Western Lightning Plus-ECL (PerkinElmer, Paris, France) and Amersham Hyperfilm (GE Healthcare).

## FLIM-FRET measurements

FLIM-FRET experiments were carried out on a HeLa$^{H2B-2FPs}$ cell line stably expressing GFP and mCherry tagged histone H2B as previously described (*Llères et al., 2009*). Fluorescence Lifetime Imaging Microscopy (FLIM) was performed using an inverted laser scanning multiphoton microscope LSM780 (Zeiss) equipped with temperature-controlled environmental black walls chamber. Measurements were acquired at 37°C, with a 63× oil immersion lens NA 1.4 Plan-Apochromat objective from Zeiss. Two-photon excitation was achieved using a Chameleon Ultra II tunable (680–1080 nm) laser (Coherent) to pump a mode-locked frequency-doubled Ti:Sapphire laser that provided sub-150-femtosecond pulses at a 80-Mhz repetition rate with an output power of 3.3 W at the peak of the tuning curve (800 nm). Enhanced detection of the emitted photons was afforded by the use of the HPM-100 module (Hamamatsu R10467-40 GaAsP hybrid PMT tube). The fluorescence lifetime imaging capability was provided by TCSPC electronics (SPC- 830; Becker & Hickl GmbH). TCSPC measures the time elapsed between laser pulses and the fluorescence photons. EGFP and mCherry fluorophores were used as a FRET pair. The optimal two-photon excitation wavelength to excite the donor (EGFP) was determined to be 890 nm (*Llères et al., 2007*). Laser power was adjusted to give a mean photon count rate of the order $4.10^4$–$10^5$ photons/s. For imaging live cells by FLIM, the standard growth medium was replaced with phenol red-free DMEM supplemented with 10% FBS. Fluorescence lifetime measurements were acquired over 90 s and fluorescence lifetimes were calculated for all pixels in the field of view (256×256 pixels) and then a particular region of interest (e.g., nucleus) was selected using SPCImage software (Becker & Hickl, GmbH).

## Analysis of the fluorescence lifetime measurements for FRET experiments

The analysis of the FLIM measurements was performed by using SPCImage software. Because FRET interactions cause a decrease in the fluorescence lifetime of the donor molecules (EGFP), the FRET efficiency can be calculated by comparing the FLIM values obtained for the EGFP donor fluorophores in the presence and absence of the mCherry acceptor fluorophores. Mean FRET efficiency images were calculated such as the FRET efficiency, $E_{FRET} = 1 - (\tau_{DA}/\tau_D)$, where $\tau_{DA}$ is the mean fluorescence lifetime of the donor (H2B-EGFP) in the presence of the acceptor (mCherry-H2B) expressed in the HeLa$^{H2B-2FPs}$ and $\tau_D$ is the mean fluorescence lifetime of the donor (H2B-EGFP) expressed in HeLa$^{H2B-GFP}$ in the absence of acceptor. In the non-FRET conditions, the mean

fluorescence lifetime value of the donor in the absence of the acceptor was calculated from a mean of the $\tau_D$ by applying an exponential decay model to fit the fluorescence lifetime decays.

In the FRET conditions, we applied a biexponential fluorescence decay model to fit the experimental decay curves $f(t) = a \, e^{-t}/\tau_{DA} + b \, e^{-t}/\tau_D$. By fixing the noninteracting proteins lifetime tD using data from control experiments (in the absence of FRET), the value of tDA was estimated. Then, the FRET efficiency ($E_{FRET}$) was derived by applying the following equation: $E_{FRET} = 1 - (\tau_{DA}/\tau_D)$ at each pixel in a selected ROI using SPCImage software. The FRET distribution curves from these ROIs were displayed from the extracted associated matrix using SPCImage and then normalized and graphically represented using Excel and GraphPad Prism software.

## Immunofluorescence

Cells were seeded on 12 mm diameter coverslips #1.5 coated with 1% gelatine. Before fixation coverslips were washed once with PBS. Then, cells were fixed either in 3.7% formaldehyde for 15 min at RT or in cold 100% MeOH (10 min, -20°C). Formaldehyde fixed cells were immediately washed twice with PBS and permabilized in 0.2% TRITON X-100 for 15 min at RT, while MeOH fixed cells on coverslips were transferred on tissue paper and kept at RT to dry. Next, cells were blocked in blocking solution (5% FBS; 0.1% Tween-20 in PBS) for 30 min at RT, incubated overnight with primary antibodies diluted in blocking solution at 4°C, washed 3 times 5 min with PBS-Tween (0.1% Tween-20 in PBS), incubated with secondary antibody at RT for 1 hr, and washed 4 times 5 min with PBS-Tween. Secondary antibodies were diluted 1:1000 for fluorophores Alexa488; 555; 568 and 1:500 for fluorophore Alexa647. Coverslips were washed in distilled water prior mounting on slide with ProLong Gold Antifade Reagent with DAPI.

## Nucleolar imaging

HeLa or U2OS cell lines were cultured in 96-well plates. For siRNA-mediated Ki-67 depletion, a transfection reagent (mix of 0.125 μl of Interferin and 20 μl of Optimem) was added to each well of the plate and left for 10 min at RT°. SiRNA (10 μl of 100 nM) were added and left for another 30 min at RT°. Cells (70 μl of 300,000 cells/ml dilution) were then added to each well and the plates were incubated for 3 days at 37°C with 5% CO2. Nucleolar structure disruption was performed by treatment of the cells with 0.2 μg/ml of Actinomycin D, 40 μM roscovitine or 60 μM DRB for 90 min.

For immunofluorescence, cells were fixed in 2% formaldehyde, washed in PBS and blocked in PBS supplemented with 5% BSA and 0.3% Triton X-100 during 1 hr at RT°. Anti-Pes 1 antibody (1:1,000; courtesy from E. Kremmer), anti-Ki-67 antibody (1:500, Cell Signaling) and/or anti-Fibrillarin antibody (1:250, antibodies online) were diluted in PBS supplemented with 1% BSA, 0.3% Triton X-100 and incubated with the cells O/N at 4°C. Cells were washed in PBS and incubated with the secondary antibody coupled to AlexaFluor 488 or 594 (1:1,000; Invitrogen) in PBS, 1% BSA, 0.3% Triton X-100 for 1 hr at RT°. Cells were washed in PBS and treated with DAPI.

Microscopy was performed on a Zeiss Axio Observer.Z1 microscope driven by MetaMorph (MDS Analytical Technologies, Canada). Images were captured in the confocal mode using a Yokogawa spindisk head and the HQ2 camera with a laser illuminator from Roper (405 nm 100 mW Vortran, 491 nm 50 mW Cobolt Calypso, and 561 nm 50 mW Cobolt Jive) and 40x or 100x objectives (Zeiss). Line scans and images were constructed using Image J. The CellProfiler software was used to quantify the DAPI intensity at the peri-nucleolar region of about 100 individual cells and classical statistical t-test was applied to the data to compare the intensity distributions.

## Immunohistochemistry

Freshly dissected small intestines were flushed and fixed for 4 hr in neutral buffered formalin before paraffin embedding. Briefly, 5-μm-thick sections were dewaxed in xylene and rehydrated in graded alcohol baths. Antigen retrieval was performed by boiling slides for 20 min in 10 mM sodium citrate buffer, pH 6.0. Nonspecific binding sites were blocked in blocking buffer (TBS, pH 7.4, 5% dried milk, and 0.5% Triton X-100) for 60 min at RT. Sections were then incubated with primary antibodies diluted in blocking buffer overnight at 4°C. Primary antibodies used were as follows: anti Ki-67 (Ab16667) and anti DCLK1 (Ab31704) were from Abcam, Cambridge, UK. Anti beta-catenin (BD610154) was from BD-Bioscience, Oxford, UK. Anti BrdU (G3G4) was form the Developmental Studies Hybridoma Bank. Slides were then washed two times with 0.1% PBS-Tween (Sigma-Aldrich)

before incubation with fluorescent secondary antibodies conjugated with either Alexa 488, Cyanin-3, or Cyanin-5 (Jackson ImmunoResearch Laboratories, Inc.) and Hoechst at 2 µg/ml (Sigma-Aldrich) in PBS–Triton X-100 0.1% (Sigma-Aldrich). Stained slides were then washed two extra times in PBS before mounting with Fluoromount (Sigma-Aldrich). Methods used for bright-field immunohistochemistry were identical, except that slides were incubated with 1.5% $H_2O_2$ in methanol for 20 min and washed in PBS to quench endogenous peroxydase activity before antigen retrieval. Envision+ (Dako) was used as a secondary reagent. Signals were developed with DAB (Sigma-Aldrich) and a hematoxylin counterstain (DiaPath) was used. After dehydration, sections were mounted in Pertex (Histolab). Goblet cells staining was achieved with a periodic acid/Schiff's reaction (Sigma-Aldrich)."

## FISH

Fish of major satellite DNA in combination with immunofluorescence was performed in formaldehyde fixed cells, as in (*Saksouk et al., 2014*).

## Table of antibodies used in western blotting and immunofluorescence

| Protein | Clone | WB dilution | IF dilution | Company | Reference |
|---|---|---|---|---|---|
| human Ki-67 | SP6 | 1/300 | 1/300 | Abcam | ab16667 |
| human Ki-67 | 35/Ki-67 | 1/200 | 1/200 | BD Transduction Laboratories | 610968 |
| mouse Ki-67 | SolA15 | 1/300 | 1/300 | Ebioscience | 14-5698-80 |
| human pRB | G3-245 | 1/300 | x | BD Pharmingen | 554136 |
| human cyclin A | 6E6 | 1/250 | 1/100 | Novocastra | NCL-CYCLIN A |
| cyclin A | H-432 | 1/500 | 1/300 | Santa Cruz Biotechnology | sc-751 |
| phospho-Histone H3 (Ser 10) | | 1/500 | | Cell signaling | 9701 |
| trimethyl-Histone H3 (Lys 9) | | x | 1/500 | Milipore | 07-442 |
| trimethyl-Histone H4 (Lys 20) | | x | 1/500 | Abcam | ab9053 |
| HP1α | 15.19s2 | x | 1/500 | Milipore | 05-689 |
| HP1β | | x | 1/200 | Abcam | ab10478 |
| HP1γ | 2MOD-1G6 | | | Active motif | 39981 |
| nucleolin | 4E2 | x | 1/500 | Abcam | ab13541 |
| CENP-A | | x | 1/400 | Cell signaling | 2186 |
| Pes1 | | 1/1000 | | Gift from E. Kremmer | |
| Fibrillarin | | 1/250 | | Antibodies online | |

## Nuclear extract preparation

### Materials

Buffer A (10 mM HEPES pH 7.9; 10 mM KCl; 0,1 mM EDTA; 0.1 mM EGTA + freshly added 0.2 mM $Na_3VO_4$; 20 µM MG132; 1 mM DTT; Complete-Protease inhibitor cocktail); IGEPAL CA-630 (Sigma-Aldrich); Buffer C (20 mM HEPES pH 7.9; 1 mM EDTA; 1 mM EGTA; 400 mM NaCl; 25% glycerol + freshly added 0.2 mM $Na_3VO_4$; 20 µM MG132; 1 mM DTT; Complete-Protease inhibitor cocktail).

$10^7$ cells were harvested by trypsinization, washed once with cold PBS, pelleted by centrifugation and resuspended in 200 µL chilled buffer A. Then, cells were incubated on ice for 5 min. 10% IGEPAL CA-630 was added to each lysate (to a final concentrationof 0.2%) and the samples were vortexed for 15 s prior to incubation on ice for 15 min. The lysates were pelleted by centrifugation at 13,000 x

g for 30 s at 4°C, supernatants were kept as a cytoplasmic fraction. Residual pellets were resuspended in 300 μL of ice-cold buffer C prior to incubation for 1 hr at 4°C with rotation. Next, the lystates were vortexed for 30 s and centrifuged for 10 min at high speed at 4°C. Supernatant were collected as a nuclear extract and stored at -80°C. Protein concentration were determined by Bradford assay (Sigma-Aldrich).

## Ki-67 mRNA engagement in polysome fractions

Cells were pre-treated 5 min with 20 μg/mL of emetine, before being collected, washed and resuspended in ice cold homogenization buffer (50 mM Tris-HCl ph7.5, 5 mM $MgCl_2$, 25 mM KCl, 0.2M Sucrose, 0.5% NP-40, EDTA-free protease inhibitors (Roche), 10 U/ML RNAse Out (Invitrogen), DEPC water). We then lyzed cells using Lysing Matrix D beads and FastPrep sample preparation system (MP Bio). The cleared lysate was layered on 15–50% sucrose gradient in the same buffer (homogenization buffer minus NP-40). Following centrifugation at 35,000 rpm (Beckman, SW41.Ti) for 2.5 hr at 4°C, gradients were fractionated (density gradient fractionator, Teledyne Isco) with absorbance measured continuously at 254 nm. We isolated RNA from fractions with TRIzol (Thermo Fisher Scientific) following the manufacturer's instructions. We then reverse transcribed purified RNA into cDNA following RT-PCR method. We analysed Ki-67 mRNA level in polysome fractions using two *Mki67* primer pairs (5'-AATCCAACTCAAGTAAACGGGG-3', 5'-TTGGCTTGCTTCCATCCTCA-3' and 5'-CATCAGCCCATGATTTTGCAAC-3', 5'-CTGCGAAGAGAGCATCCATC-3') normalizing to housekeeping genes (*Gapdh*: 5'-AAATGGTGAAGGTCGGTGTG-3', 5'-AATCTCCACTTTGCCAC TGC-3'; *B2m*: 5'-GGTCTTTCTGGTGCTTGTCT-3', 5'-GCAGTTCAGTATGTTCGGCTT-3'; *Actb*: 5'-TCCTGGCCTCACTGTCCAC-3', 5'-GTCCGCCTAGAAGCACTTGC-3'; *Hprt*: 5'-AAGCCTAAGA TGAGCGCAAG-3', 5'-TTACTAGGCAGATGGCCACA-3').

## Cloning of cDNA of full-length Ki-67 tagged with GFP and 3xFLAG
### Materials
SuperScript II Reverse Transcriptase (ThermoFisher), Pfu DNA Polymerase (Promega, Lyon, France), pGEM-T Easy Vector (Promega), Gateway pENTR Vector (LifeTechnologies), KpnI (NEB), AlfII (NEB), Ligase T4 (Promega).

### Table of primers

| Name | Sequence |
| --- | --- |
| BamHI-AF | AAGGATCCGCCGCCACCATGTGGCCCACGAGACGCCTG |
| AR | GGCCACGTGCCGTGTCTTTCA |
| BF | AAGGAGCAACCGCAGTT |
| BR | TGTGTCCATAGCTTTCCCTAC |
| CF | GATAAAGGCATCAACGTGTTC |
| CR | GGAGTTTATGAAGCCGATTC |

Total RNA purified from exponentially growing HeLa cells was used as a template in reverse transcription reaction using SuperScript II Reverse Transcriptase (ThermoFisher) with primers AR, BR and CR to synthesize cDNA template. Full length Ki-67 cDNA was obtained by cloning into pGEM-T vector of three overlapping parts amplified using Pfu DNA Polymerase and primers pairs BamHI-AF/AR, BF/BR and CF/CR. Three parts of full length cDNA were joined together by digestion and ligation reactions. Parts B and C were digested with KpnI enzyme and ligated together. Parts A and BC were digested with AlfII and ligated together. Full length cDNA were cloned into Gateway pENTR Vector containing KOZAK sequence by digesting with BamHI introduced site in 5' site and with SacII in 3' site. Next, KOZAK-Ki-67 cDNA were transferred into Gateway pDEST vector (pcDNA5/GFP/3xFLAG/FRT) to tagged Ki-67 construct with GFP and 3xFLAG sequence at 5' site.

## Immunoprecipitation of 3xFLAG Ki-67

1.5x10$^7$ U2OS cells were transfected with pcDNA5 plasmid expressing 3xFLAG-Ki-67.

As a control, an equal number of cells were transfected with pcDNA3 plasmid expressing 3xFLAG-TRIM39 or 3xFLAG. 24 hr after transfection, cells were harvested and the nuclear extracts prepared. 100 µg of nuclear protein extract were combined with 40 µl anti-FLAG M2– agarose beads (Sigma-Aldrich), and incubated for 1 hr at 4°C with rotation. Beads were washed 5 times for 5 min at 4°C with rotation with washing buffer (20 mM HEPES pH 7.9; 1 mM EDTA; 1 mM EGTA; 150 mM NaCl; 25% glycerol + freshly added 0.2 mM Na$_3$VO$_4$; Complete-Protease inhibitor cocktail) and the precipitates were eluted by 50 µL of SDS denaturation buffer, and heating at 95°C for 5 min.

## Mass spectrometry analysis

Eluted proteins were reduced, alkylated, analysed in a 4–20% gradient gel (BioRad) and entire lanes were sliced. Tryptic peptides were prepared for mass spectrometry, essentially as described, and then concentrated with a pre-column (Thermo Scientific, C18 PepMap100, 300 µm × 5 mm, 5 µm, 100 A) at a flow rate of 20 µL/min using 0.1% formic acid. Samples were separated with a C18 reversed-phase capillary column (Thermo Scientific, C18 PepMap100, 75 µm × 250 mm, 3 µm, 100 A) at a flow rate of 0.3 µL/min using the following gradient: 8–28% acetonitrile in 40 min and then from 28–42% in 10 min. The HPLC system was coupled online to a Q-TOF Maxis Impact (Bruker Daltonik GmbH, Bremen, Germany) mass spectrometer. Up to 30 data-dependent MS/MS spectra were acquired in positive ion mode. MS/MS raw data were analysed using Data Analysis software (Bruker Daltonik GmbH, Bremen, Germany) to generate the peak lists. The Homo sapiens Complete Proteome database (downloaded on Uniprokb 20131108, contains 88,266 sequences) was queried locally using the Mascot search engine v.2.2.07 (Matrix Science, http://www.matrixscience.com) and with the following parameters: 2 missed cleavages, carbamidomethylation of Cysteine as fixed modification and oxidation of Methionine, phosphorylation of Threonine and Serine as variable modifications. MS tolerance was set to 20ppm for parent ions and 0.05 Da for fragment ions.

For SILAC, samples were prepared as described (*Skorupa et al., 2013*). Peptides were analysed online by nano-flow HPLC–nanoelectrospray ionization using an Q Exactive mass spectrometer (Thermo Fisher Scientific) coupled to an Ultimate 3000 RSLC (Dionex, Thermo Fisher Scientific). Desalting and pre-concentration of samples were performed on-line on a Pepmap pre-column (0.3 mm × 10 mm, Dionex). A gradient consisting of 0–40% B in A for 60 min, followed by 80% B/ 20% A for 15 min (A = 0.1% formic acid, 2% acetonitrile in water; B = 0.1% formic acid in acetonitrile) at 300 nL/min was used to elute peptides from the capillary reverse-phase column (0.075 mm × 150 mm, Pepmap, Dionex).

Raw data analysis was performed using the MaxQuant software (v. 1.5.0.0) (*Cox and Mann, 2008*) using standard parameters except Requantity option set as TRUE or FALSE. Peak lists were searched against the UniProt Mouse database (release 2015_11; http://www.uniprot.org), 255 frequently observed contaminants as well as reversed sequences of all entries.

Graphical representations were generated using perseus (1.5.3.2).

## RT-PCR

### Materials

10 mM dNTPs (LifeTechnologies), 50 µM random hexaprimers (NEB, Evry, France), SuperScript II Reverse Transcriptase (ThermoFisher), RNasin Plus RNase Inhibitor (Promega).

1000 ng of purified RNA in total volume of 10 µL, extracted by RNeasy Mini Kit (Qiagen, Paris, France), were mixed with 1 µL of 10 mM dNTPs (2.5 mM of each) and 1 µL of 50 µM random hexaprimers (New England Biolabs). Samples were incubated at 65°C for 5 min, then immediately transferred on ice. Next, into samples were added 5 µL of 5xFirst Strand Buffer, 2 µL 100 mM DTT and 1 µL RNasin RNase Inhibitor. Samples were incubated at 25°C for 10 min and at 42°C for 2 min. 1 µL of SuperScript II Reverse Transcriptase was added to each sample, prior to incubation at 42°C for 60 min, 70°C for 15 min.

## qPCR

qPCR was performed using LightCycler 480 SYBR Green I Master (Roche, Grenoble, France) and LightCycler 480 qPCR machine. The reaction contained 5 ng of cDNA, 2 µL of 1 µM qPCR primer

pair (final concentration of each primer was 200 nM in reaction mixture), 5 µL 2x Master Mix, and final volume made up to 10 µL with DNase free water. qPCR was conducted at 95°C for 10 min, and then 40 cycles of 95°C for 20 s, 58°C for 20 s and 72°C for 20 s. The specificity of the reaction was verified by melt curve analysis. Each sample was performed in three replicates.

## Table of qPCR primers (Tm - 60°C)

| Target | Forward | Reverse |
|---|---|---|
| human *MKI67* Qiagen | Qiagen QuantiTect Hs_MKI67_1_SG | |
| human *MKI67* | TGACCCTGATGAGAAAGCTCAA | CCCTGAGCAACACTGTCTTTT |
| human *CCNA2* | AGGAAAACTTCAGCTTGTGGG | CACAAACTCTGCTACTTCTGGG |
| human *CCNE1* | CCGGTATATGGCGACACAAG | ACATACGCAAACTGGTGCAA |
| human *CCNB1* | TGTGTCAGGCTTTCTCTGATG | TTGGTCTGACTGCTTGCTCT |
| human *CDC6* | TTGCTCAGGAGATTTGTCAGG | GCTGTCCAGTTGATCCATCTC |
| human *FZR1* | TCTCAGTGGAAGGGGACTCA | CAACATGGACAGCTTCTTCCC |
| human *B2M* (norm) | GCGCTACTCTCTCTTTCTGG | AGAAAGACCAGTCCTTGCTGA |
| human *RPL19* (norm) | ATGCCGGAAAAACACCTTGG | GTGACCTTCTCTGGCATTCG |
| mouse *Mki67* | AATCCAACTCAAGTAAACGGGG | TTGGCTTGCTTCCATCCTCA |
| mouse *B2M* (norm.) | GGTCTTTCTGGTGCTTGTCT | GCAGTTCAGTATGTTCGGCTT |

## RNA electrophoresis

For analysis of high-molecular-weight species, 5 µg of total RNA were resolved on agarose denaturing gels (6% formaldehyde/1.2% agarose in HEPES-EDTA buffer). For the analysis of the low-molecular-weight RNA species 5 µg of total RNA were separated on denaturing acrylamide gels (8% acrylamide-bisacrylamide 19:1/8 M urea in Tris-borate-EDTA buffer [TBE]) for 4 hr at 350 V.

## Northern blotting

Agarose gels were transferred by capillarity overnight in 10× saline sodium citrate (SSC) and acrylamide gels by electrotransfer in 0.5× TBE on nylon membranes (GE Healthcare). Membranes were prehybridized for 1 hr at 65°C in 50% formamide, 5× SSPE, 5× Denhardt's solution, 1% w/v SDS, 200 µg/ml fish sperm DNA solution (Roche). The 32P-labeled oligonucleotide probe was added and incubated for 1 hr at 65°C and then overnight at 37°C.

## Sequences of the probes

| Oligo probe name | Sequence |
|---|---|
| LD1827 (ITS1) | CCTCGCCCTCCGGGCTCCGGGCTCCGTTAATGATC |
| LD1828 (ITS2) | CTGCGAGGGAACCCCCAGCCGCGCA |
| LD1829 | GCGCGACGGCGGACGACACCGCGGCGTC |
| LD1844 (5'-ETS) | CGGAGGCCCAACCTCTCCGACGACAGGTCGCCAGAGGACAGCGTGTCAGC |
| LD2079 (5'-ITS2) | GGGGCGATTGATCGGCAAGCGACGCTC |
| LD2132 (5.8S mat) | CAATGTGTCCTGCAATTCAC |
| LD2133 (7SL) | GCTCCGTTTCCGACCTGGGCC |
| LD2655 | GGAGCGGAGTCCGCGGTG |

## TALEN-mediated Ki-67 mutant mice

Plasmids encoding two TALEN pairs were purchased from Cellectis (Paris, France). They were designed to bind the following sequence of *Mki67* gene: 5' <u>TCCCGACGGCCGGGCGG</u> ACCATGGCGTCCTC <u>GGCTCACCTGGTCACCA</u> 3. Underlined sequences are recognised by the left or the right TALEN, respectively. Plasmids were linearized by PacI digestion and used as a template forin vitro transcription to produce TALEN-encoding mRNAs using T7 RiboMAX Express System (Promega). Transcripts were purified using MEGAclear Transcription Clean-Up Kit (Ambion, Thermo-Fisher). Quality and quantity of transcribed mRNAs were verified by BioAnalyzer (Agilent, Paris, France). Next, 32 ng or 8 ng of each TALEN-encoding mRNAs were injected into zygotes and implanted into 36 or 18 mice, respectively. 7 chimeric mutant mice were obtained with deletions ranging between 1nt and 38 nt by injection of 32 ng of each mRNAs (22% NHEJ) and 1 chimeric mouse with deletion of 24 nt by injection of 8 ng of each mRNAs. Founder mice were crossed and we obtained four mice for germline transmission (1nt, 2nt, 3nt, 24nt deletion).

## Genotyping of TALEN-mediated Ki-67 mutant mice

Genomic DNA was purified from mouse-tail piece using KAPA Express Extract Kit (Kapa Biosystems, London, UK). PCR was conducted using the primers 5'GGCCAGAGCTAACTTGCGCTGACTG 3 'and 5'AAACAGGCAGGAGCTGAGGCTCAGC 3 'and Pfu DNA Polymerase (Promega). Product size 203 bp. Then, PCR product was cleaned up using ExoSAP-IT (Affymetrix, High Wycombe, UK) and sequenced using 5'GGCCAGAGCTAACTTGCGCTGACTG 3 'primer.

## Histology of TALEN-mediated Ki-67 mutant mice

Genotyped mice pups were fixed in 4% paraformaldehyde for 48 hr and formol 10% 3 days prior to after longitudinal section in 2 parts. Embryos were decalcified in EDTA 10% - Formol 2,5% before paraffin embedding. Tissue was dehydrated through a series of graded ethanol baths and then infiltrated with wax. Infiltrated tissues were then embedded into wax blocks. From these blocks, 5-μm-thick sections were cut and then stained with hematoxylin.

## MEF isolation

MEF were isolated from E13.5 embryos of the corresponding genotype. The female was killed by cervical dislocation. The uterine horns were dissected and placed into a petri dish containing PBS. Each embryo was separated from its placenta and surrounding membranes. The brain, all dark red organs and the intestine were cut away and blood was removed as much as possible. The remaining parts of the embryo were transferred into a dish containing 1 ml of 1x Trypsin-EDTA 0.25%. They were finely minced with a razor blade and incubated at 37°C for 1 hr in a 5% $CO_2$ incubator. Trypsin was inactivated with 4 ml of DMEM supplemented with 10% fetal bovine serum and 1% Penicillin/Streptomycin and the carcass was homogenized by several passages up and down using a pipet. Finally, 6 ml of DMEM media were added and cells were incubated at 37°C in an incubator containing 3% $O_2$ and 5% $CO_2$.

## Immunohistochemistry

Mouse intestinal epithelium was processed for immunohistochemistry as described (*Gerbe et al., 2011*). Cdh1 knockout mice were analysed by immunohistochemistry as described (*Eguren et al., 2013*).

## Single TALEN-mediated Ki-67 KO in NIH-3T3 mouse fibroblasts

NIH-3T3 cells were plated at a density of $3x10^4/cm^2$ in the afternoon the day before transfection. Cells were transfected with plasmids encoding: TALEN pair targeted to initial ATG described in section TALEN-mediated Ki-67 KO mouse and pEGFP, or pEGFP by itself. Next day, eGFP positive cells were sorted by FACS (BD FACSAria) and around 240 cells from each condition were plated in five 96-well plates (480 wells). Two weeks later, we obtained around 50 clones from each condition. Then, TALEN-mediated mutants were screened for Ki-67 expression by immunofluorescence. Nine selected clones were then screened by PCR and sequencing.

## Sequencing and PCR analysis of single TALEN Ki-67 mutant NIH-3T3 cell lines

Genomic DNA were purified from harvested cells using KAPA Express Extract Kit (Kapa Biosystems). PCR product was amplified using the primers 5' AGAGCTAACTTGCGCTGACT 3' and 5' TCGCGTC TACCGAGTGTAAAA 3' and Pfu DNA Polymerase (Promega). Product size 364 bp. For sequencing one additional amplification cycle was performed using Taq Polymerase to add a 3' dA overhang on the end of PCR fragment. Next, PCR product was ligated with pGEM-T Easy Vector (Promega). Competent bacteria were transformed with ligation reaction and plated on agar plates with ampicillin, IPTG and X-Gal. Next day ten white colonies were selected from each individual ligation reaction to perform plasmid preparation. Purified plasmids were sequenced using T7 and SP6 RNA Polymerase transcription initiation site primers.

## Generation of TALEN pair targeted to site of STOP codon of the mouse *MKI67* gene

TALENs were designed using TAL Effector Nucleotide Targeter 2.0 (Cornell University) software to bind following sequence of *Mki67* gene: 5' <u>TACCAGAAAAGTGAAA</u>CTATGTAGCAAA<u>GACATTTAA-GAAGGAAAAGT</u> 3' and assembled using The Golden Gate TALEN kit (AddGene). Underlined sequences are recognised by the left TALEN or the right TALEN, respectively.

## Double TALEN-mediated eGFP transgenic Ki-67 mutant

NIH-3T3 cells were plated at density of $3 \times 10^4 / cm^2$ in the afternoon the day before transfection. Cells were transfected with plasmids encoding: TALEN pair targeted to initial ATG described in section TALEN-mediated Ki-67 KO mouse; TALEN pair targeted to site of STOP codon MKI67 gene; reporter system (*Kim et al., 2013*) containing STOP codon area as a target sequence; linearized construct containing *Mki67* locus replaced by eGFP gene. Two days after transfection, hygromycin selection was performed by culturing the cells in the presence of 2 mg/ml of hygromycin B for two days at 37°C. For clonal analysis, around 500 hygromycin-selected cells were plated in ten 96-well plates (960 wells). Two weeks after, around 100 clones (10% ) were screened by immunofluorescence for Ki-67 expression.

## DNA replication assay

Analysis of DNA replication progress in cells was achieved by treatment with 10 µM 5-ethynyl-2'-deoxyuridine (EdU) (ThermoFisher) before fixation. Replicating cells were visualized following the protocol from Click-iT EdU Alexa Fluor 488 Imaging Kit (ThermoFisher).

## RNA synthesis assay

Analysis of newly synthesised RNA in cells was achieved by treatment with 2 mM 5-ethynyl uridine (EU) (ThermoFisher) for 20 min before fixation. Replicating cells were visualized following the protocol from Click-iT EU Alexa Fluor 488 Imaging Kit (ThermoFisher).

## Flow cytometry

### Cell cycle analysis

Cells were harvested, washed once with cold PBS, then, resuspended in 300 µL cold PBS and fixed with 700 µL chilled 100% methanol. Cells were kept at -20°C up to one day of analysis, but at least overnight. On the day of analysis, cells were pelleted by centrifugation at 6000 rpm for 5 min. After washing once with 1% BSA in PBS, cells were stained with Propidium Iodide staining solution (10 µg/ml Propidium Iodide, 1% BSA, 200 µg/ml RNase A in PBS) for 15 min at room temperature and subjected to cell cycle analysis using BD FACS Calibur (BD Biosciences, SanJose, CA).

### DNA replication assay - EdU labelling

Cells were harvested, washed once with cold PBS, then, resuspended in 300 µL cold PBS and fixed with 700 µL chilled 100% methanol. Cells were kept at -20°C up to the day of analysis, but at least overnight. On the day of analysis, cells were pelleted by centrifugation at 6000 rpm for 5 min. After washing once with 1% BSA in PBS cells were stained with Click-iT EdU Alexa Fluor 488 Flow

Cytometry Assay Kit (ThermoFisher). The Click-iT TM EdU Flow Cytometry Assay system (Invitrogen) was used following the manufacturer's instructions.

## Microarray analysis - transcriptome

RNA was prepared using RNeasy Mini Kit (Qiagen) following the manufacturer's instructions from U2OS shRNA non-targeting CTRL, U2OS shRNA Ki-67, HeLa shRNA non-targeting control CTRL, HeLa shRNA Ki-67 grown 3 months after initial infection with lentivirus armed pGIPZ shRNA. RNA was purified from three shRNA CTRL tumour xenografts or three shRNA Ki-67 tumour xenografts isolated from mouse C1-SM (33 days after injection), C1-OD (41 days after injection) and C2-2OR (46 days after injection) using TRIzol reagent (Life Technologies) following the manufacturer's instructions. Cy3-labelled cRNA was amplified and hybridized on the Agilent SurePrint G3 Human GE 8x60k Microarray according to the procedures by Imaxio company (Lyon, France). Raw data were preprocessed using GeneSpring GX software (Agilent Technologies) to define differently expressing genes and present data by clustered heat-maps.

## Statistical analysis of transcriptome

Significant differences between experimental groups were determined using an unpaired two-tailed Student t test in Prism 5 (GraphPad). For all analyses, p values <0,05 (*), p values < 0,01 (**) and p values <0,001 (***) were considered statistically significant. Transcripts that (i) demonstrated at least a 1,5-fold change in expression, (ii) had a greater-than-background signal intensity value and were determined to be 'Present' by Affymetrix algorithms, and (iii) had a value that was significant by Student's t test and FDR (Benjamini Hochberg) correction (p<0.02 (U2OS, HeLa); p<0.2 (Xenografts)) were considered differentially expressed.

## Acknowledgements

Many thanks to all members of the Fisher lab for helpful discussions and criticism of the paper, and all technical staff of MRI imaging facility, RHEM histology facility and IGMM mouse facility. Thanks to Daniel Gerlich, Thierry Forné, Chris Lord for helpful discussions and for reading the manuscript.

## Additional information

### Funding

| Funder | Grant reference number | Author |
|---|---|---|
| Agence Nationale de la Recherche | ANR-09-BLAN-0252 | Nikolaos Parisis Daniel Fisher |
| Ligue Contre le Cancer | EL2013.LNCC/DF | Daniel Fisher |
| Ligue Contre le Cancer | Graduate student fellowship | Michal Sobecki |
| Fondation pour la Recherche Médicale | Graduate student fellowship | Michal Sobecki |
| GEFLUC Languedoc Roussillon | Subvention | Daniel Fisher |

The funders had no role in study design, data collection and interpretation, or the decision to submit the work for publication.

### Author contributions

MS, NP, Conception and design, Acquisition of data, Analysis and interpretation of data, Drafting or revising the article; KM, SU, Acquisition of data, Analysis and interpretation of data, Drafting or revising the article; AC, EN, DL, FG, AD, Conception and design, Acquisition of data, Analysis and interpretation of data; SP, LK, Acquisition of data, Drafting or revising the article; ME, Acquisition of data, Contributed unpublished essential data or reagents; M-CB, Conception and design, Acquisition of data, Contributed unpublished essential data or reagents; SH, Acquisition of data, Analysis and interpretation of data; JD, PJ, DLJL, DF, Conception and design, Analysis and interpretation of data, Drafting or revising the article; MM, Drafting or revising the article, Contributed unpublished

essential data or reagents; VD, Analysis and interpretation of data, Drafting or revising the article; RF, Conception and design, Drafting or revising the article

### Author ORCIDs

Nikolaos Parisis, http://orcid.org/0000-0002-5706-0122
Manuel Eguren, http://orcid.org/0000-0003-0850-939X
Marcos Malumbres, http://orcid.org/0000-0002-0829-6315
Robert Feil, http://orcid.org/0000-0002-5671-5860
Daniel Fisher, http://orcid.org/0000-0002-0822-3482

### Ethics

Animal experimentation: All animal experiments were performed in accordance with international ethics standards and were subjected to approval by the Animal Experimentation Ethics Committee of Languedoc Roussillon and the Ministry for Higher Education and Research

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
