## [Decision Letter]

Thank you for submitting your work entitled "Ki-67 organises heterochromatin during cell proliferation" for consideration by *eLife*. Your article has been reviewed by two peer reviewers, and the evaluation has been overseen by a Reviewing Editor and Fiona Watt as the Senior Editor.

The reviewers have discussed the reviews with one another and the Reviewing Editor has drafted this decision to help you prepare a revised submission.

Summary:

The manuscript has improved and is now more focused on revisiting the prevailing models for Ki-67 functions and heterochromatin organization. Considering that the work represents a paradigm shift in our understanding of the role of Ki-67, it is likely of broad interest to *eLife* readers. The major concerns raised by reviewers have been acceptably answered: assessment of remaining Ki-67 expression in the Ki-67 mutants is exhaustive and well performed; all subtypes of heterochromatin protein (HP1alpha/β/γ) were tested. However, the reviewers have some outstanding concerns.

Essential revisions:

One remaining concern is with the H3K9 and K4K20 data that leads the authors to conclude "trimethylation of histone H3K9 and H4K20 at heterochromatin is strongly reduced" in the abstract. Looking at the data, in Figure 10—figure supplement 1, this is not apparent. It does seem reduced, but it is hard to tell. The authors could do some quantification here as they have done for some of the other microscopy-based data. A result featured so strongly in the Abstract should be backed by clear data. Either the authors should tone down the Abstract and put the caveats in the paper, or bolster these data.

[Editors’ note: a previous version of this study was rejected after peer review, but the authors submitted for reconsideration. The previous decision letter after peer review is shown below.]

Thank you for choosing to send your work, "Ki-67 organises heterochromatin to control gene expression during cell proliferation", for consideration at *eLife*. Your work has been seen by three reviewers, one of who is a member of the Board of Reviewing Editors. Although the work is of interest, we regret to inform you that the findings at this stage are too preliminary for further consideration at *eLife* as they currently stand.

In principle all three reviewers were interested in your discoveries. We think that the role of Ki-67 in vivo is a potentially important advance.

However we have two main concerns.

Firstly, the paper came across as a bit incoherent. We think that this is mainly due to the large number of different models and techniques you used, which could be addressed by streamlining the writing of the paper. I think that this aspect could be improved through changes to the text to allow them to streamline this and frame things on a key question or a couple of main points. Alternatively, you could eliminate some of the unnecessary approaches from the paper to assist this streamlining. Generally, rather than setting the paper up as opposing previous work in some way, the authors should know that this paper clearly stands out as an important advance on its own. Highlighting their work for what it reveals related to the core functions of Ki-67 and citing the other work where appropriate for both the similarities and differences will nicely indicate this.

Secondly, and more importantly, there are a number of technical problems that we feel preclude publication at this point.

We are generally worried about your mouse alleles.

1) The way in which the authors generated these mice was through the introduction of TALEN-mediated mutations. These would disrupt aspects of Ki-67 expression based on the mutations listed in Figure 4, but it is certainly possible that there is the expression of a N-terminally truncated product utilizing a downstream AUG. The highly cropped Western blot presented in this figure does not exclude the possibility that there are other versions of the protein produced that confer some function. It is also not clear from their Methods section which region of the protein the utilized antibodies recognize, and whether it is possible that alternative forms of this are produced in their mutants.

2) The fact that tissues from the Ki-67 delete mouse still stains for Ki-67 is concerning. The authors suggest that this could be due to a downstream ATG that is still functional. This should be investigated using RT-qPCR.

At a minimum, we would like you to more thoroughly investigate the state of your alleles. But this could mean generating new mouse alleles, which would clearly require a new submission because it would take longer than two months.

We also have some other technical issues.

1) Previous studies that identified important roles for Ki-67 in nucleolar structure and RNA biogenesis (most recently the paper by Booth et al.) used siRNA-mediated knockdowns, and in fact they rescued the phenotypes that they observed using hardened constructs. For the experiments conducted in this paper in human cells, they primarily used shRNA-mediated depletion, which is typically less efficient.

2) rRNA biogenesis does appear to be affected at some steps (e.g. 47S). Quantification and replicates are necessary.

3) There appears to be a modest cell cycle defect after Ki-67 disruption (see Figures 3E, Figure 2—figure supplement 1B, Figure 2—figure supplement 2C). The cell cycle certainly isn't halted, however, it would be good to soften claims that use strong wording like, "dispensible". To support the conclusion, single cells have to be filmed for around the doubling time (or 24 hr or longer) upon knockdown or knockout of Ki-67.

4) While the authors describe "Surprisingly, HP1 localization was not altered (Figure 7B)" in the Results section, they tested only HP1beta. HP1 has subtypes of α, β, and γ, with HP1a and g relatively more important in cell division. I am very surprised at this over-simplification. HP1alpha and HP1gamma must be also stained in the background of Ki-67 downregulation. Without this experiment, any statement about "HP1 localization" is difficult to make with confidence.

We realize that this is a high bar to set-but we think that sorting out these technical issues is important for your work to have the impact that it deserves. This is especially true in this case because of the varied role of KI-67 reported in the literature. Please note that we aim to publish articles with a single round of revision that would typically be accomplished within two months.

---

## [Author Response]

Essential revisions: One remaining concern is with the H3K9 and K4K20 data that leads the authors to conclude "trimethylation of histone H3K9 and H4K20 at heterochromatin is strongly reduced" in the abstract. Looking at the data, in Figure 10—figure supplement 1, this is not apparent. It does seem reduced, but it is hard to tell. The authors could do some quantification here as they have done for some of the other microscopy-based data. A result featured so strongly in the Abstract should be backed by clear data. Either the authors should tone down the Abstract and put the caveats in the paper, or bolster these data.

We would argue that the statement in the Abstract was technically correct – that H3K9 and H4K20 is strongly reduced at heterochromatin (though it is not changed in overall levels). For example, in Figure 10A, the H3K9me3 no longer colocalises with the intense DAPI spots that identify heterochromatin. However, we accept that this could be slightly misleading if read wrongly, and it is true that this is hard to see if one does not look closely, especially for Figure 10—figure supplement 1. And of course there remained a doubt as to whether heterochromatin organisation itself is the same in the knockdown cells (which it probably isn’t, in spite of the retention of HP1 staining).

We have thus made the following changes.

1) We changed the wording in the Abstract to: “Trimethylation of histone H3K9 and H4K20 was relocalised within the nucleus”.

2) We have used imaging tools to better represent the H3K9 and H4K20 staining in U2OS and HeLa cells control and knockdown cells (new Figure 10—figure supplements 1 and 2). Using the “Fire” Look-up table pseudocolour scheme, the altered patterns of histone methylation become more apparent, especially as we have enlarged these images. This shows that on Ki-67 knockdown, the majority of cells show less intense peak methylation mark staining, and both the peak intensity of the mark and the DAPI-dense chromatin are relocalised away from the nucleoli, confirming a relocalisation of H3K9me3/H4K20me3, and possibly of heterochromatin itself.

3) We have quantified the number of cells showing each pattern, and this is presented graphically below the images (new Figure 10—figure supplements 1 and 2).

Unfortunately, we could not perform the same exercise for the BJ-hTERT cells, where nucleoli (and the histone methylation marks) are much less visible. We could not objectively define nucleoli in these cells, and our attempts at co-staining H3K9me3/H4K20me3 (which is best visualised using methanol fixation) and nucleolin (best visualised using formaldehyde fixation) were not successful. Therefore, we kept the original qualitative BJ—hTERT data, since they show a qualitatively similar result in non-transformed cells, and separated it into a new figure supplement: Figure 10—figure supplement 3.

[Editors’ note: the author responses to the first round of peer review follow.]

*In principle all three reviewers were interested in your discoveries. We think that the role of Ki-67 in vivo is a potentially important advance. However we have two main concerns.*

*Firstly, the paper came across as a bit incoherent. We think that this is mainly due to the large number of different models and techniques you used, which could be addressed by streamlining the writing of the paper. I think that this aspect could be improved through changes to the text to allow them to streamline this and frame things on a key question or a couple of main points. Alternatively, you could eliminate some of the unnecessary approaches from the paper to assist this streamlining. Generally, rather than setting the paper up as opposing previous work in some way, the authors should know that this paper clearly stands out as an important advance on its own. Highlighting their work for what it reveals related to the core functions of Ki-67 and citing the other work where appropriate for both the similarities and differences will nicely indicate this.*

This is a very good suggestion. It was a highly multidisciplinary study of several aspects of Ki-67 function and regulation with several conclusions. It is indeed possible to focus on the key points. We have now done this in three main ways, and we agree that the result is a much more streamlined paper:

1) We have completely reordered the paper. We now start with the key in vivo results of overexpression and mutation of Ki-67 in the mouse. We then follow on with the characterisation of cell proliferation in the functional knockout mouse lines. All complementary work on cell proliferation in human cancer cell lines using RNAi has been relegated to supplementary information. We continue with the proteomics of Ki-67 interactors that opens up avenues in ribosome biogenesis and chromatin regulation. To take this further, we present work on rRNA processing and gene expression. The rest of the paper is then dedicated to the role of Ki-67 in organising heterochromatin.

2) We have removed two somewhat more descriptive sections. These sections were on control of Ki-67 expression on CDK4/6 activity and proteasome-mediated degradation, and on requirements for Ki-67 for tumour progression using xenografts of knockdown and control cancer cells in mice. We considered that these were potentially important from a medical point of view and so included them in the original paper. But thanks to the criticisms of reviewers we have now decided it would be more appropriate to remove these and publish them in a more specialised journal.

3) We have strengthened the existing sections on the cell cycle, ribosome biogenesis and control of heterochromatin with additional work, detailed below.

*Secondly, and more importantly, there are a number of technical problems that we feel preclude publication at this point. We are generally worried about your mouse alleles. 1) The way in which the authors generated these mice was through the introduction of TALEN-mediated mutations. These would disrupt aspects of Ki-67 expression based on the mutations listed in Figure 4, but it is certainly possible that there is the expression of a N-terminally truncated product utilizing a downstream AUG. The highly cropped Western blot presented in this figure does not exclude the possibility that there are other versions of the protein produced that confer some function. It is also not clear from their Methods section which region of the protein the utilized antibodies recognize, and whether it is possible that alternative forms of this are produced in their mutants.*

*2) The fact that tissues from the Ki-67 delete mouse still stains for Ki-67 is concerning. The authors suggest that this could be due to a downstream ATG that is still functional. This should be investigated using RT-qPCR. At a minimum, we would like you to more thoroughly investigate the state of your alleles. But this could mean generating new mouse alleles, which would clearly require a new submission because it would take longer than two months.* This is also a perfectly valid comment. We consider that it is vitally important that researchers in genetics, and mouse genetics in particular, should be aware that endonuclease-generated “knockouts” may not be functional knockouts. Many researchers these days (including ourselves) target genes in cell lines using CrispR to ablate the translation initiation ATG codon. We did this for Ki-67 in both cell lines and in the mouse using the related (and still recent) technology of TALENs. Because Ki-67 antibodies are highly sensitive (they recognise a highly repeated sequence and have been extensively optimised due to their biomedical use), we can detect what appears to be residual expression in biallelic mutants, both in the mouse and in monoclonal cell lines. Typically, we see many smaller lower intensity bands even in wild-type cells, probably resulting either from protein degradation (even within cells) or alternative splicing isoforms. All bands are strongly reduced on knockdown or mutation, but we do not usually blot the entire membrane. Western blots already presented were not artificially cropped, the membranes had been cut to allow probing for other proteins at different molecular weights. We agree that it is important to investigate further the apparent residual expression in Ki-67 mutants and we have spent the majority of the last three months doing this. Briefly, we have done the following:

1) We have revitalised a second mouse line that we had frozen as sperm. Whereas the first (∆2nt) mouse retained the ATG translation initiation codon but had an immediate downstream frameshift, the second mouse line (∆21nt) has lost the initiation ATG codon. We investigated whether this would generate a functional knockout and if not, how this would affect expression. ∆21nt homozygotes show an identical lack of phenotype to the first mouse, and a very similar residual level of Ki-67 protein in proliferating tissues, that we estimated to be around 10%. Additionally, we isolated MEFs from the ∆2nt mouse and performed qRT-PCR and Western blotting. We present an uncropped Western blot demonstrating that there is no appearance of a shorter form. This led to an identical conclusion: destroying the ATG eliminates the majority, but probably not all, of the expression.

2) We have characterised the residual expression by exploiting two new monoclonal mouse cell lines that we generated, which have similar genetic alterations to the ∆2nt and ∆21nt mice, respectively. Importantly, by Western blotting and immunofluorescence these showed very similar residual Ki-67 expression levels to the mutant mice and MEFs. The use of immortal mouse cell lines rather than proliferative tissues (the intestine is, of course, sensitive to protein degradation) enabled us to undertake a sophisticated analysis of translation. This involved, firstly, qRT-PCR of mRNA levels, showing that mRNA expression is not decreased (there is no nonsense-mediated decay). We confirmed that this is also true in the intestinal tissue and in MEFs. We then performed two series of experiments that are technically virtually impossible in vivo. In the first, we purified and quantified Ki-67 mRNA from different ribosomal fractions. Again, neither mutant showed a decrease in levels that would have indicated a decrease in translation initiation. This implies that the approximately 10-fold reduction seen in protein levels by Western blotting results not from defective translation initiation but from frameshifted (and potentially truncated) translation. Finally, we performed SILAC-proteomics by mass spectrometry to quantify Ki-67 protein in the mutants and attempt to identify whether it is truncated or not. This showed that in both mutants Ki-67 is undetectable – no Ki-67 heavy peptides were identifiable by MS/MS. Nevertheless, by expected peak m/z isotopic pattern similarity to the control sample peptides, we could identify and quantify a number of peptides in the mutants that may derive from Ki-67. This led to several main conclusions. Firstly, there is probably residual Ki-67 translation in the biallelic mutants but the level is *extremely* low. No residual expression could be positively identified in either of the two mutants, one of which retains the ATG on one allele but has an immediate frameshift thereafter, the other lacking the ATG altogether. Quantitation of the putative Ki-67 peptides in the mutants revealed that this protein is present at about 10% of the level of full length Ki-67 in the WT. Finally, no putative peptides were detected that could derive from the N-terminus, suggesting that the basal expression is of an N-terminally truncated protein.

3) We have investigated by Southern blotting the nature of the alleles in the TALEN mutant clones that have a total absence of Ki-67 expression. This shows a multiple tandem insertion of the GFP gene upstream of the Ki-67 initiation ATG.

*We also have some other technical issues. 1) Previous studies that identified important roles for Ki-67 in nucleolar structure and RNA biogenesis (most recently the paper by Booth et al.) used siRNA-mediated knockdowns, and in fact they rescued the phenotypes that they observed using hardened constructs. For the experiments conducted in this paper in human cells, they primarily used shRNA-mediated depletion, which is typically less efficient.*

We mainly used shRNA to confirm in human cells results seen using mutation of the Ki-67 gene in the mouse. Our shRNA was, however, highly optimised to ensure the most effective knockdown possible. We firstly characterised three different shRNA vectors and chose the most efficient. We then used increasingly high levels of puromycin selection after lentiviral infection, and FACs sorting based on intensity of the co-expressed turbo-GFP or RFP, to generate stable cell lines with the highest expression. For inducible shRNA we additionally used high doxycyclin levels for induction of the shRNA. However, the experiment using inducible shRNA in non-transformed cells to prevent Ki-67 expression upon re-entry into the cell cycle shows less effective knockdown. This is because the shRNA is not well induced by doxycyclin in serum-starved cells. We were only able to do this experiment by inducing the shRNA prior to serum starvation. We should add, however, that we also used siRNA for several sets of experiments, notably the H2B-FRET and pre-rRNA processing experiments. Nobody, however, has performed rescue of full-length Ki-67 using si/sh-resistant constructs: careful reading of the Booth et al. paper shows that the siRNA-resistant Ki-67 employed is only a fairly short N-terminal fragment (that nevertheless rescued the nucleolar protein targeting).

*2) rRNA biogenesis does appear to be affected at some steps (e.g. 47S). Quantification and replicates are necessary.*

There indeed does appear to be a very slight increase in the 47S product; we have verified and quantified this in replicates in several different cell lines.

*3) There appears to be a modest cell cycle defect after Ki-67 disruption (see Figure 3E, Figure 2—figure supplement 1B, Figure 2—figure supplement 2C). The cell cycle certainly isn't halted, however, it would be good to soften claims that use strong wording like, "dispensible". To support the conclusion, single cells have to be filmed for around the doubling time (or 24 hr or longer) upon knockdown or knockout of Ki-67.*

We originally showed 5-day growth curves that show that there is no significant decrease in cell proliferation in the mutants. As requested, we have now done time-lapse on the mutant cells that express no residual Ki-67. We thus quantified cell cycle length. Within a clone there is significant variability in cell cycle length between cells. While one of the two mutant clones (60) has a marginally longer cell cycle (p<0.05), this appears to be a clone effect as the other clone (65) does not. The latter clone does, however, show an unusual mitotic chromosome morphology, as the Booth et al., paper also shows using siRNA. Cells could, however, divide normally. And we have quantified EdU incorporation by FACs to determine the sensitivity of the mutant clones to serum starvation. This shows that, indeed, there is a mild effect of loss of Ki-67 in these mutant clones: cells starve more effectively, fewer of them remain able to incorporate EdU after 72h at 0.1% serum. Nevertheless, this does not alter our conclusion that starved cells are perfectly capable of entering into and progressing through the cell cycle in the absence of detectable Ki-67. We have, as requested, altered the language used in the paper.

*4) While the authors describe "Surprisingly, HP1 localization was not altered (Figure 7B)" in the Results section, they tested only HP1beta. HP1 has subtypes of α, β, and γ, with HP1a and g relatively more important in cell division. I am very surprised at this over-simplification. HP1alpha and HP1gamma must be also stained in the background of Ki-67 downregulation. Without this experiment, any statement about "HP1 localization" is difficult to make with confidence.*

We had actually at various times performed localisation of all three HP1 isoforms in different cell lines, but not always in the same experiment. We did not see any obvious effect of Ki-67 loss on localisation pattern of any isoform. We have now done this systematically to look at all three isoforms in the same experiment, and we reproduce our previous conclusions.